# Equivalence of Context and Parameter Updates in Modern Transformer Blocks

**Adrian Goldwaser** [1 2]   **Michael Munn** [2]   **Javier Gonzalvo** [2]   **Benoit Dherin** [2]

## Abstract

Recent research has established that the impact of context in a vanilla transformer can be represented implicitly by forming a token-dependent, rank-1 patch to its MLP weights. This work extends that foundational theory to the diverse architectures of modern Large Language Models. We first demonstrate a precise, analytical solution for a Gemma-style transformer block, proving that the entire effect of a context can be perfectly mapped to rank-1 patches on its MLP weight matrices and a patch to the RMSNorm scale. We then generalize this result, providing a constructive proof and algorithm for multi-layer models. To unify these findings, we introduce a general framework centered on two core properties: *input controllability* and *output controllability*. We prove that a perfect implicit weight patch is possible for any MLP block where the inner function is input-controllable and the outer function is output-controllable. This provides a simpler and more powerful lens for understanding how transformer models transmute prompts into effective weights. This setup generalizes to a wide range of modern LLM architectures including gating, pre-/post-norm, mixture of experts and sequential/parallel transformer blocks.

## 1. Introduction

Large Language Models (LLMs) exhibit a remarkable, almost paradoxical, capability: after their large-scale training is complete, they appear to learn new tasks and adapt their behavior "on the fly" based purely on the prompt they are given. This powerful emergent phenomenon, known as in-context learning (ICL) (Brown et al., 2020), is a central mystery. How does a static, pre-trained network effectively "reprogram" itself at inference time? One emerging perspective is that the model doesn't just process the context; it absorbs it. Recent research has begun to formalize this idea, showing that the prompt can be mathematically re-interpreted as a set of implicit, task-specific modifications—or patches—to the model's own weights (Dai et al., 2023; Dherin et al., 2025).

This line of inquiry was given a precise, mechanistic foundation by Dherin et al. (2025), who proved that for a single, vanilla transformer block (Vaswani et al., 2017), the computational effect of a context is mathematically equivalent to a specific, rank-1 patch to the MLP weight matrix and bias vector.

However, this proof was developed for a vanilla transformer block, leaving open the question of its applicability to the complex and varied architectures used in modern models. These models employ different components, such as gated MLPs (e.g., SwiGLU, GeGLU) (Shazeer, 2020) which utilize activation functions like GELU (Hendrycks and Gimpel, 2016), RMSNorm (Zhang and Sennrich, 2019), and Pre-Normalization schemes (Xiong et al., 2020) and do not use biases, which were required to absorb the impact of the residual connection. The compatibility of these modern architectures with the implicit weight patch mechanism has not been formally analyzed.

This paper bridges that gap and provides a comprehensive, general theory for implicit weight updates in modern transformers. Our main contributions are:

- We provide a constructive proof for a modern, Gemma-style architecture, deriving the exact parameter patches required to perfectly absorb context into the MLP and normalization layers (**Theorem 1** in Section 3.1).

- We extend this finding inductively to deep, multi-layer models, proving that a perfect patch exists for the entire network (**Theorem 2** in Section 3.2) and provide a practical algorithm for its computation (Algorithm 1).

- We introduce a general framework built on two core properties, *input controllability* (Definition 1) and *output controllability* (Definition 2). We use this framework to prove a **unified theorem (Theorem 3)** that generalizes our findings to a wide range of architec-

[1]University of Cambridge, UK [2]Google Research. Correspondence to: Adrian Goldwaser <goldwaser@google.com>, Michael Munn <munn@google.com>, Benoit Dherin <dherin@google.com>.

*Proceedings of the 43rd International Conference on Machine Learning*, Seoul, South Korea. PMLR 306, 2026. Copyright 2026 by the author(s).

tures including Gemma, Llama, Falcon, Mistral, and MoE models (Section 5).

- We experimentally validate our theory on a Gemma 3 model for both text and image contexts and a Falcon model for text contexts, showing that the patched model without context achieves near-perfect logit matching and identical token generation to the original model with context (Section 4).

**Conflict of Interest Disclosure.** The authors M.M., J.G., and B.D. are employed by Google, which leads the development of Gemma, one of the models evaluated in this paper. A.G. is affiliated with both the University of Cambridge and Google Research.

## 2. Background

The mechanism driving the in-context learning (ICL) phenomenon observed by Brown et al. (2020) remains a central research question. Several complementary theories have emerged to explain how transformers adapt at inference time. Some frame ICL as a high-level form of implicit Bayesian inference, where the model uses the prompt to update an internal belief state (Xie et al., 2022). Others have proposed that the transformer's forward pass is mathematically analogous to an optimization process, effectively performing steps of gradient descent on an implicit objective defined by the context (von Oswald et al., 2023). At a more mechanistic level, ICL capabilities have also been linked to the emergence of specific "induction heads" during training, which allow the model to perform pattern-matching and copying (Olsson et al., 2022).

Our analysis builds upon the more recent and granular work of Dherin et al. (2025), which formalizes how a transformer block processes context. Their framework centers on the *contextual block*: a contextual layer (like self-attention or a state space model (Gu et al., 2022; Gu and Dao, 2023)) followed by a neural network, typically an MLP. The key insight is that the influence of the context can be viewed as an implicit patch to the weights of the MLP.

In a vanilla transformer block, the output of the attention layer is added to the input via a residual connection. We will refer to this resulting vector $\mathbf{v}$ without a specific context, and $\mathbf{v}_C$ with it. The change induced by the context is thus $\Delta \mathbf{v} = \mathbf{v}_C - \mathbf{v}$. This vector $\mathbf{v}_C$ is then passed through an MLP, which in simple models concludes with a final bias addition, for instance, $f(\mathbf{z}) = W_2 \cdot \text{act}(W_1 \mathbf{z} + \mathbf{b}_1) + \mathbf{b}_2$ (Vaswani et al., 2017). Dherin et al. (2025) proved that the effect of processing $\mathbf{v}_C$ is mathematically identical to processing the original vector $\mathbf{v}$ with a modified MLP. In this modified MLP block, the entire contextual difference is absorbed by a rank-1 patch to the input weight matrix, $W_1$ and a patch to the bias ($\Delta \mathbf{b}_2 = \Delta \mathbf{v}$). This elegant solution, however,

critically depends on the existence of the $\mathbf{b}_2$ term. Modern high-performance models like Gemma (Kamath et al., 2025) and Llama (Touvron et al., 2023) have eliminated these biases, leaving a gap in the theory. This theory also needs to be extended to multi-layer networks, and blocks with normalization layers and gating. Our work fills in these missing pieces to extend this setup to modern architectures.

Innocenti and Achour (2025) independently explored generalizations for Pre-LayerNorm and arbitrary sequence/block positions, though their analysis is restricted to standard residual blocks containing biases and acknowledges a lack of exact correspondence to practical model architectures. In contrast, we introduce a unified controllability framework that encompasses the bias-free, gated architectures and RMS normalization used in state-of-the-art models. Furthermore, while previous work discusses iterative application to arbitrary blocks, we provide a formal inductive proof and algorithm ensuring mathematical equivalence across all layers of deep networks during autoregressive generation, specifically addressing the numerical stability required for real-world deployment.

Concurrently, other research has addressed the limitation that these implicit patches are *token-dependent* and must be recomputed at each generation step (Mazzawi et al., 2025). Mazzawi et al. (2025) proposes a method to aggregate these transient patches into a reusable, token-independent "thought patch." Our work is complementary to this direction. We do not focus on the re-usability of the patches, but on the more fundamental question of their existence and form in modern architectures. We demonstrate that the implicit weight patch is a general principle, not an artifact of vanilla transformer models, and begin by providing a constructive proof for a Gemma block.

## 3. Results: Implicit Updates in Modern Transformers

We first prove that a perfect implicit weight patch exists for a modern transformer block, specifically one modeled after the Gemma architecture. We then extend this finding to multi-layer models.

### 3.1. The Gemma Block

A standard decoder-only transformer block, such as in Gemma (Kamath et al., 2025), utilizes a pre/post-normalization architecture for its MLP sub-layer as shown in Figure 1. We analyze the processing of a single token representation, $\mathbf{x}$, given a preceding context $C$. The context $C$ consists of all the preceding tokens in the sequence.

The process begins with the attention block. We define the function $A(C, \mathbf{x}) = \mathbf{x} + \text{Attn}(C, \mathbf{x})$ to represent the entire attention sub-layer's operation: it computes the attention

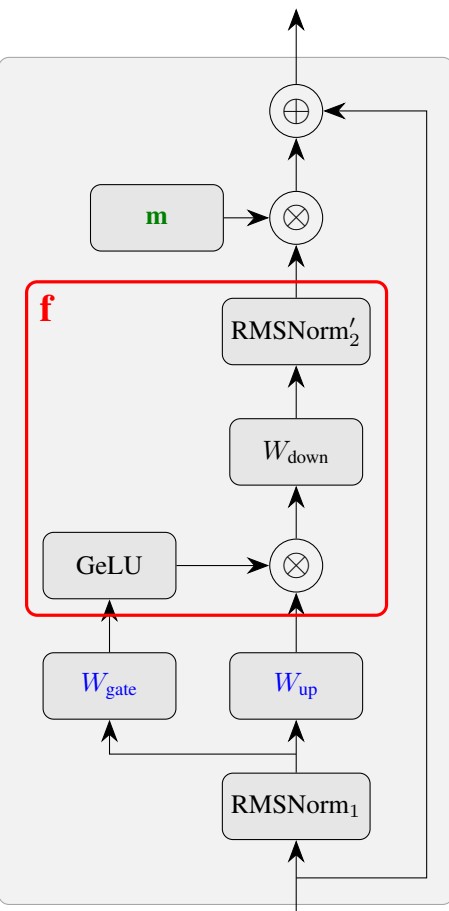

*Figure 1.* Gemma MLP block diagram. $\mathbf{m}$ is part of the second RMS normalization (RMSNorm$_2$) but stated separately to match the equations. $\otimes$ denotes elementwise multiplication of vectors.

output (for an input $\mathbf{x}$ with context $C$) and adds it back to the original input $\mathbf{x}$ via the residual connection.

Our analysis centers on comparing the block's computation with this full context $C$ against its computation with a *reduced context*, $C \setminus Y$, where $Y$ represents the "extra" contextual information we wish to absorb (e.g., the in-context learning examples). We will focus on the case where the reduced context is empty, i.e., $Y = C$, which means $C \setminus Y = \emptyset$.

We first define the forward pass using the full context $C$. Let the intermediate result from the attention sub-layer, which serves as the input to the MLP sub-layer, be denoted by $\mathbf{v}_C$:

$$\mathbf{v}_C = A(C, \mathbf{x})$$

This vector is then normalized using RMSNorm (Zhang and Sennrich, 2019) before being processed. Let the normalized vector be $\mathbf{z}_C$:

$$\mathbf{z}_C = N_{\text{RMS}}(\mathbf{v}_C)$$

The MLP sub-layer's output is scaled and added back to $\mathbf{v}_C$ in the MLP's residual connection. The complete output of

the transformer block, $T(C, \mathbf{x})$, is given by:

$$T(C, \mathbf{x}) = \mathbf{v}_C + \mathbf{m} \odot f(W_{\text{gate}}\mathbf{z}_C, W_{\text{up}}\mathbf{z}_C) \qquad (1)$$

where $f$ is the MLP's core computation (e.g., GELU activation and element-wise multiplication (Shazeer, 2020)), for Gemma, $f(\mathbf{a}, \mathbf{b}) = \text{RMSNorm}'(W_{\text{down}}(\text{GeLU}(\mathbf{a}) \odot \mathbf{b}))$. $W_{\text{gate}}$ and $W_{\text{up}}$ are trainable weight matrices, and $\mathbf{m}$ is a trainable output scaling vector included in the RMS normalization. We use RMSNorm$'$ above and in Figure 1 to mean RMSNorm without applying the scaling vector.

**Theorem 1** (Single Block Equivalence). *Let* $\mathbf{v}_C = A(C, \mathbf{x})$ *and* $\mathbf{v} = A(C \setminus Y, \mathbf{x})$ *be the intermediate outputs from the attention sub-layer with the full context and a reduced context, respectively. Let their normalized versions be* $\mathbf{z}_C = N_{RMS}(\mathbf{v}_C)$ *and* $\mathbf{z} = N_{RMS}(\mathbf{v})$.

*The output of the transformer block with full context, $T(C, \mathbf{x})$, can be perfectly replicated using the reduced context in a modified transformer block, $T'(C \setminus Y, \mathbf{x})$, if the MLP parameters $(W_{gate}, W_{up}, \mathbf{m})$ are adjusted by the following updates and $f(W_{gate}\mathbf{z}_C, W_{up}\mathbf{z}_C)_i \neq 0$ for all $i$:*

$$\Delta W_{gate} = \frac{W_{gate}(\mathbf{z}_C - \mathbf{z})\,\mathbf{z}^\top}{\|\mathbf{z}\|^2} \qquad (2)$$

$$\Delta W_{up} = \frac{W_{up}(\mathbf{z}_C - \mathbf{z})\,\mathbf{z}^\top}{\|\mathbf{z}\|^2} \qquad (3)$$

$$\Delta \mathbf{m} = (\mathbf{v}_C - \mathbf{v}) \oslash (f(W_{gate}\mathbf{z}_C, W_{up}\mathbf{z}_C)) \qquad (4)$$

*where the division $\oslash$ in (4) is performed element-wise.*

*Proof.* We show the equivalence by substituting the parameter updates into the definition of the modified transformer block's output, $T'(C \setminus Y, \mathbf{x})$.

First, observe that the update $\Delta W_{\text{gate}}$ is constructed to align the MLP's internal state. The input to the gate projection becomes:

$$(W_{\text{gate}} + \Delta W_{\text{gate}})\mathbf{z} = W_{\text{gate}}\mathbf{z} + \frac{W_{\text{gate}}(\mathbf{z}_C - \mathbf{z})\mathbf{z}^\top \mathbf{z}}{\|\mathbf{z}\|^2} = W_{\text{gate}}\mathbf{z}_C.$$

An identical result holds for $W_{\text{up}}$. This ensures the internal MLP activation (post normalization but pre-scaling) is unchanged, i.e., $f((W_{\text{gate}} + \Delta W_{\text{gate}})\mathbf{z}, (W_{\text{up}} + \Delta W_{\text{up}})\mathbf{z}) = f(W_{\text{gate}}\mathbf{z}_C, W_{\text{up}}\mathbf{z}_C) \equiv \mathbf{h}_{\text{mlp}}$.

Substituting this result and the definition of $\Delta \mathbf{m}$ into the output equation for $T'$ yields:

$$\begin{aligned}
T'(C \setminus Y, \mathbf{x}) &= \mathbf{v} + (\mathbf{m} + \Delta \mathbf{m}) \odot \mathbf{h}_{\text{mlp}} \\
&= \mathbf{v} + (\mathbf{m} \odot \mathbf{h}_{\text{mlp}}) + (\Delta \mathbf{m} \odot \mathbf{h}_{\text{mlp}}) \\
&= \mathbf{v} + (\mathbf{m} \odot \mathbf{h}_{\text{mlp}}) + \left(\frac{\mathbf{v}_C - \mathbf{v}}{\mathbf{h}_{\text{mlp}}}\right) \odot \mathbf{h}_{\text{mlp}} \\
&= \mathbf{v}_C + \mathbf{m} \odot \mathbf{h}_{\text{mlp}} = T(C, \mathbf{x}). \qquad \square
\end{aligned}$$

where the division is taken elementwise.

**Note.** *The logic of this proof rests on a separation of concerns. The rank-1 updates to $W_{gate}$ and $W_{up}$ are designed to counteract the change in the* normalized *input vector ($\mathbf{z}$ versus $\mathbf{z}_C$), ensuring the internal MLP computation remains identical (input controllability). The subsequent update to the output scale $\mathbf{m}$ then perfectly absorbs the difference from the* pre-normalization *residual path ($\mathbf{v}$ versus $\mathbf{v}_C$, output controllability), guaranteeing the final output is exactly equal, i.e. $T'(\mathbf{x}) = T(C, \mathbf{x})$.*

**Note.** *This update is mathematically correct, but numerically unstable when used with lower precision datatypes, we show this in Section 4 and address this in Appendix B.*

### 3.2. Extension to Multi-Layer Architectures

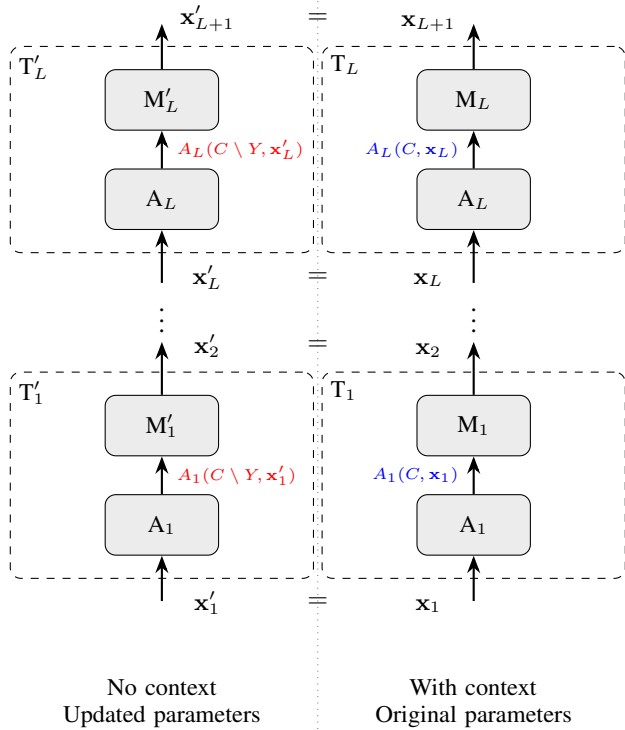

No context
Updated parameters

With context
Original parameters

*Figure 2.* Multi-layer equivalence diagram. The left column shows the model with updated parameters and no explicit context. The right column shows the original model with full context. At each layer $i$, we have $\mathbf{x}'_{i+1} = T'_i(C \setminus Y, \mathbf{x}'_i) = T_i(C, \mathbf{x}_i) = \mathbf{x}_{i+1}$. The deltas are now $\Delta A_{\mathbf{x}_i}(Y) = A_i(C, \mathbf{x}_i) - A_i(C \setminus Y, \mathbf{x}_i)$ and the equivalent normed version. Note that the $\mathbf{x}'_i$ are different from the intermediate values when simply running a forward pass with the original parameters without context.

This result can be extended from a single transformer block to a full, L-layer transformer.

**Theorem 2** (Multi-Layer Equivalence). *For an L-layer transformer where each transformer block is structured like the Gemma transformer block in Theorem 1 with parameters $\Theta = \{\theta_1, \ldots, \theta_L\}$, a sequence of updated parameters*

$\Theta' = \{\theta'_1, \ldots, \theta'_L\}$ *exists such that $T_{\Theta'}(C \setminus Y, \mathbf{x}_1) = T_{\Theta}(C, \mathbf{x}_1)$, where $T_{\Theta}$ is the full transformer with parameters $\Theta$, assuming the conditions for Theorem 1 are satisfied at every layer.*

*Proof.* We proceed by induction on the layer index $k$, from 1 to $L$.

For the base case ($k = 1$), the input is the token embedding $\mathbf{x}_1$, which is identical for both the original model (with full context $C$) and the updated model (with reduced context $C \setminus Y$). Per Theorem 1, we can therefore find a parameter update $\theta'_1$ for the first transformer block such that its output $\mathbf{x}'_2 = T_1(C \setminus Y, \mathbf{x}_1; \theta'_1)$ is identical to the original output $\mathbf{x}_2 = T_1(C, \mathbf{x}_1; \theta_1)$.

Now, assume for a layer $k > 1$ that we have chosen updates $\{\theta'_1, \ldots, \theta'_{k-1}\}$ such that the input to the $k$-th transformer block is identical in both forward passes, i.e., $\mathbf{x}'_k = \mathbf{x}_k$. The outputs of this transformer block are $\mathbf{x}_{k+1} = T_k(C, \mathbf{x}_k; \theta_k)$ for the original model and $\mathbf{x}'_{k+1} = T_k(C \setminus Y, \mathbf{x}_k; \theta'_k)$ for the updated one.

Since the transformer block input $\mathbf{x}_k$ is equal for both, the conditions of Theorem 1 are met. Thus, an update $\theta'_k$ exists that makes the outputs identical, ensuring $\mathbf{x}'_{k+1} = \mathbf{x}_{k+1}$.

By the principle of induction, this procedure can be applied sequentially for all layers $k = 1, \ldots, L$. Each step preserves the hidden state, guaranteeing that the final output of the updated model with reduced context is identical to the original model's output. $\square$

### 3.3. Algorithm for Multi-Layer Updates

The inductive proof of Theorem 2 gives rise to a practical, layer-by-layer algorithm for computing the required weight updates for the entire model. The key is to first perform a forward pass with the full context to record the target activations at each layer. Then, in a second sequence of passes, we compute the updates for each layer sequentially. To compute the update at a layer, we use the output of the previous layer with the previous patch applied, this makes the algorithm self-correcting in the presence of numerical errors.

It is theoretically possible to fully incorporate the context using just the final layer, however this runs into practical issues in real-world scenarios. See Appendix C for experiments investigating this.

## 4. Experimental Validation

To validate our theoretical findings, we perform experiments to confirm that our computed weight updates perfectly replicate the behavior of standard in-context learning. Specifically, we test the hypothesis that the updated model, op-

**Prompt:** Write a single-sentence weather forecast for Mars, from the perspective of a slightly annoyed robot:

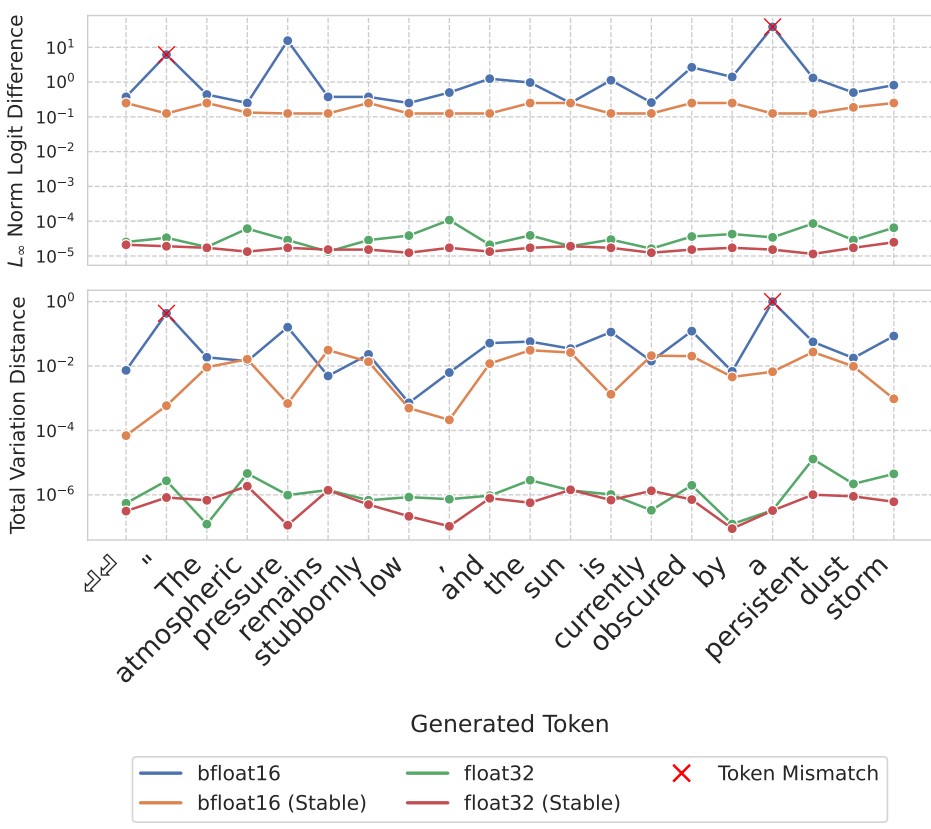

*Figure 3.* **Comparison of generation metrics between the original and updated models.** The top plot shows the $L_\infty$ norm of the logit difference and the bottom plot shows the Total Variation Distance plotted at each step of the token generation process. The x-axis displays the sequence of generated tokens. We show this separately for each platform/data type. A red 'X' indicates that the predicted tokens did not match there.

erating without context, is functionally equivalent to the original model with context. This equivalence is evaluated by comparing their output probability distributions on a token-by-token basis during generation.

### 4.1. Setup

We use instruction-tuned Gemma 3 1B (Instruction Tuned) and 4B models. The task is to generate a fictional forecast based on a specific instructional prompt. The prompt, which serves as the context $C$ for the initial update, is: "Write a single-sentence weather forecast for Mars, from the perspective of a slightly annoyed robot:". Other prompts are shown in Appendix C.

The experiment compares two scenarios:

1. **Baseline (with Context):** Generate the forecast using the original model, conditioned on the full prompt $C$.

2. **Updated Model (No Context):** The forecast is generated autoregressively. For each new token, we compute a new set of updated weights $\Theta'$ using Algorithm 1. *This update absorbs the initial prompt $C$ as well as all tokens generated in previous steps.* The next token is then sampled from the model with these recomputed weights, without providing *any* explicit context.

To isolate the effect of the updates at each step, we compare the logit distributions for the next token given an identical generation history. If the models' top token choices diverge, we record the difference but force the updated model to proceed with the baseline's chosen token for all subsequent steps, allowing us to continue comparing the distributions on the same sequence. If we would ever divide by zero due to the conditions not being satisfied, we instead divide by 1 to avoid any intermediate NaN elements.

**Algorithm 1** Compute Multi-Layer Implicit Weight Updates
___

**Require:** Model parameters $\Theta = \{\theta_1, \ldots, \theta_L\}$, input $\mathbf{x}$, full context $C$, reduced context $C \setminus Y$.
1: **// Step 1: Record target activations with full context**
2: Let $\mathbf{x}_1 = \text{Embed}(C, \mathbf{x})$.
3: Initialize list of target activations $T = []$.
4: **for** $l = 1$ to $L$ **do**
5:     $\mathbf{x}_{l+1} = \text{Block}_l(\mathbf{x}_l; \theta_l)$.
6:     Append $\mathbf{x}_{l+1}$ to $T$.
7: **end for**
8: **// Step 2: Compute updates layer by layer**
9: Let $\mathbf{x}_1' = \text{Embed}(C \setminus Y, \mathbf{x})$.
10: Initialize list of updated parameters $\Theta' = []$.
11: **for** $l = 1$ to $L$ **do**
12:     Let $\mathbf{x}_{l+1,\text{target}} = T[l]$.
13:     // Compute update $\Delta\theta_l$ needed for $\text{Block}_l(\mathbf{x}_l'; \theta_l + \Delta\theta_l)$ to output $\mathbf{x}_{l+1,\text{target}}$ via Eq. 2-4 (Theorem 1) or Appendix B.
14:     $\Delta\theta_l = \text{ComputeSingleBlockUpdate}(\mathbf{x}_l', \mathbf{x}_{l+1,\text{target}}, \theta_l)$.
15:     $\theta_l' = \theta_l + \Delta\theta_l$.
16:     Append $\theta_l'$ to $\Theta'$.
17:     // The next layer's input is the output from the current layer to absorb any numerical errors.
18:     $\mathbf{x}_{l+1}' = \text{Block}_l(\mathbf{x}_l'; \theta_l')$.
19: **end for**
20: **return** $\Theta'$.
___

### 4.2. Results

Our objective is to verify that the influence of the context has been fully compiled into the updated model's weights. We hypothesize that the updated model, without context, will perfectly replicate the generation of the original model with context. To test this, we compare the models' outputs at each step of the generation process for a given context. Note that we re-calculate the updated parameters for each token.

Because the transformer equations are highly underdetermined, infinitely many updates can enforce output equivalence. Our algorithm targets specific formulations based on three desiderata:

- **Rank-1 patches:** The most minimal modification to a weight matrix, making them highly amenable to mechanistic interpretability.
- **Layer distribution:** reduces the update norm at any single layer (see Appendix C.4), again implying minimality.
- **Matrix vs vector specificity:** Matrix updates are input-conditional, altering outputs only along specific directional vectors. This, as well as numerical stability, motivates our constrained RMSNorm inversion (Appendix B) to absorb context into matrices rather than

the global scaling vector $\mathbf{m}$.

To show the universality of this update, we run experiments with both textual and image contexts. We can see the results for textual contexts in Figures 3 and 4 and image contexts in Figure 5. We can see that the logit diffs remain extremely small with `float32` maintaining perfect token matching.

To demonstrate architectural universality, we replicated our evaluation suite on the Falcon architecture. We find that our controllability updates are equally effective here, yielding near-identical output distributions to the contextualized baseline with perfect token matching even for `bfloat16`. Full metric comparisons and the complete set of generation graphs for Falcon are provided in Appendix C.3.

The comparison is based on the following metrics:

- **Token-Level Matching:** A direct check to ensure both models sample the identical token at each step using argmax generation.
- $L_\infty$ **Norm of Logits:** The maximum absolute distance between the output logits, quantifying the difference in their raw predictions.
- **Total Variation Distance:** A measure of similarity between the full probability distributions over the vocabulary. $\text{TVD}(p, q) = \frac{1}{2}\|p - q\|_1$

The results for these metrics, shown in Figure 3, confirm our hypothesis in the case of `float32`. As we move towards real-world scenarios such as `bfloat16`, they diverge slightly due to numerical precision. Even in the least accurate setup (`bfloat16`), we get 87.5% agreement on token predictions, swapping to the numerically stable version pushes this up to 100%, on par with `float32`. The `bfloat16 (Stable)` uses a more numerically stable update described in Appendix B. We show other metrics in Appendix C.

We can see that the runs with `float32` are almost exact. The update $\Delta\mathbf{m} = \frac{\mathbf{v}_C - \mathbf{v}}{f(W_{\text{gate}}\mathbf{z}_C, W_{\text{up}}\mathbf{z}_C)}$ involves element-wise division by potentially very small (or 0) numbers. This results in the extreme sensitivity to data type. We can mitigate this by using a more numerically stable update that reduces the need to update $\mathbf{m}$, but we cannot eliminate it as the high dimensionality combined with the low precision of `bfloat16` result in these numerical issues. For Falcon, the results are less sensitive to numerical issues due to the lack of post-norm and this is not required.

## 5. A General Framework for Implicit Updates

The specific results for the Gemma architecture can be generalized by introducing two fundamental properties of functions within a transformer block. Let $A(C, \mathbf{x})$ be the output of a contextual layer (e.g., attention) for input $\mathbf{x}$ and context

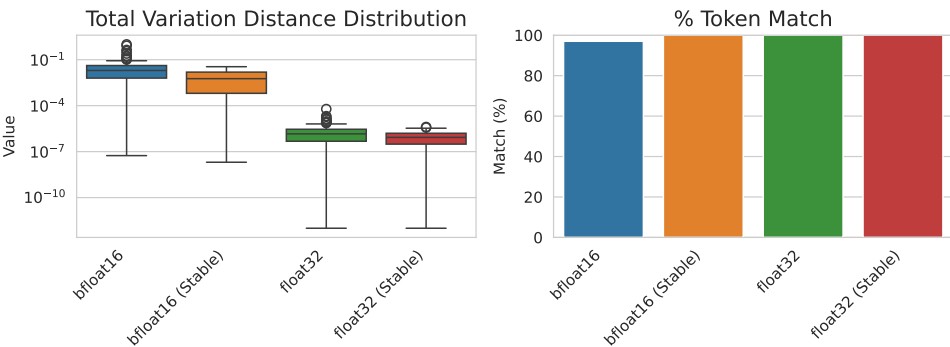

*Figure 4.* **Comparison of update accuracy for different data types and updates.** Here we show the distribution of the logit difference and the accuracy percentage over five textual generations.

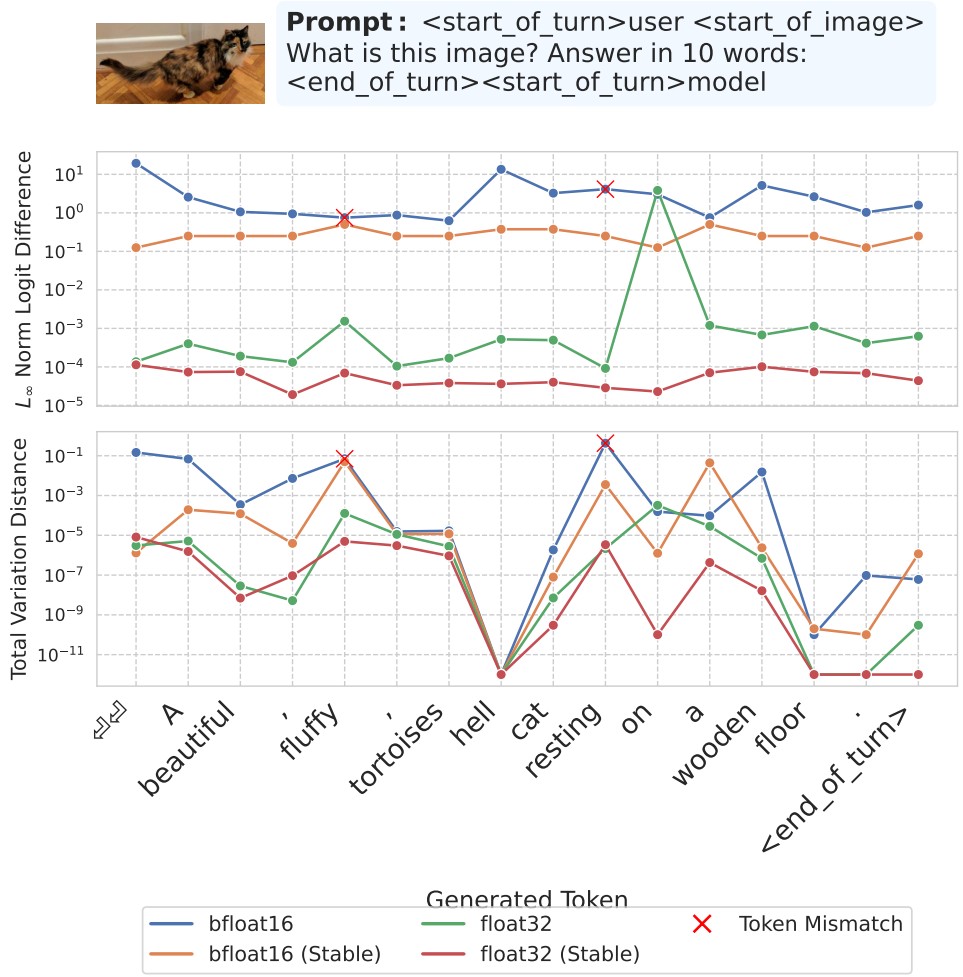

*Figure 5.* **Comparison of generation metrics between the original and updated models on images.** This is a matching experiment as Figure 3 but on Gemma 3 4B with an image as part of the context. We can see that this method continues to work with multi-modal input.

$C$. We can then define the *contextual difference vector* for an input $\mathbf{x}$ and context $Y$ as:

$$\Delta A_{\mathbf{x}}(Y) = A(C, \mathbf{x}) - A(C \setminus Y, \mathbf{x})$$

| Component / Condition | Notes | Required Update |
| --- | --- | --- |
| **Input Update for MLP** (Lemma 1) | For each input matrix $W_i$ multiplied by $\mathbf{v}_C$ (or $\mathbf{v}$ for no context). | $\Delta W_i = \dfrac{W_i(\mathbf{v}_C - \mathbf{v})\mathbf{v}^\top}{\|\mathbf{v}\|^2}$ |
| **Input Update with Pre-Normalization** (Lemma 2) | Let $\mathbf{z} = N(\mathbf{v})$ and $\mathbf{z}_C = N(\mathbf{v}_C)$. This is the update for each input matrix $W_i$. | $\Delta W_i = \dfrac{W_i(\mathbf{z}_C - \mathbf{z})\mathbf{z}^\top}{\|\mathbf{z}\|^2}$ |
| **Output Update: Outer Bias** (Lemma 3) | Covers the bias term in post-LayerNorm ($\boldsymbol{\beta}$). | $\Delta \mathbf{b} = \Delta A_\mathbf{x}(Y)$ |
| **Output Update: Outer Weight Matrix** (Lemma 4) | Covers Llama (Touvron et al., 2023), Falcon (Almazrouei et al., 2023) and others. $\mathbf{y} = f(\mathbf{v})$ is the pre-multiply output. | $\Delta W' = \dfrac{\Delta A_\mathbf{x}(Y)\mathbf{y}^\top}{\|\mathbf{y}\|^2}$ |
| **Output Update: Elementwise Multiply** (Lemma 5) | Covers the learnable scale $\mathbf{m}$ in post-RMSNorm. $\mathbf{h} = f(\mathbf{v})$ is the pre-scale output. | $\Delta \mathbf{m} = \Delta A_\mathbf{x}(Y) \oslash \mathbf{h}$ |
| **Mixture of Experts (MoE)** (Lemma 6) | Applies to Mixtral (Jiang et al., 2024). $S$ is the sum of the router gates. | Update each active expert $j$'s output params to add $\Delta A_\mathbf{x}(Y)/S$. |
| **Parallel Transformer Blocks** (Lemma 7) | Applies to GPT-J (Wang and Komatsuzaki, 2021). The MLP branch is context-independent. | Update MLP output params to absorb the entire attention change, $\Delta A_\mathbf{x}(Y)$. |

*Table 1.* Analysis of common architectural forms and their corresponding weight updates under the controllability framework.

This vector represents the entire change induced by the context $Y$ in the attention sub-layer's output.

**Definition 1** (Input Controllability)**.** *A function $f(\mathbf{z}; \boldsymbol{\theta}_f)$ is **input-controllable** if for any non-zero input vectors $\mathbf{z}$ and $\mathbf{z} + \Delta\mathbf{z}$, there exists a parameter update $\Delta_\mathbf{z}\boldsymbol{\theta}_f$ such that $f(\mathbf{z}+\Delta\mathbf{z}; \boldsymbol{\theta}_f) = f(\mathbf{z}; \boldsymbol{\theta}_f + \Delta_\mathbf{z}\boldsymbol{\theta}_f)$. This update $\Delta_\mathbf{z}\boldsymbol{\theta}_f$ may depend on $\mathbf{z}$, $\Delta\mathbf{z}$ and $\boldsymbol{\theta}_f$.*

**Definition 2** (Output Controllability)**.** *A function $g(\mathbf{v}; \boldsymbol{\theta}_g)$ is **output-controllable** if for any fixed non-zero input $\mathbf{v}$ and any desired difference vector $\Delta\mathbf{y}$, there exists a parameter update $\Delta_\mathbf{v}\boldsymbol{\theta}_g$ such that $g(\mathbf{v}; \boldsymbol{\theta}_g) + \Delta\mathbf{y} = g(\mathbf{v}; \boldsymbol{\theta}_g + \Delta_\mathbf{v}\boldsymbol{\theta}_g)$. This update $\Delta_\mathbf{v}\boldsymbol{\theta}_g$ may depend on $\mathbf{v}$, $\Delta\mathbf{y}$ and $\boldsymbol{\theta}_g$.*

With these definitions, we can state a simpler, more general theorem for implicit weight updates, for which the Gemma-specific theorem is a special case.

**Theorem 3** (Unified Theorem for Residual Blocks)**.** *For a residual MLP block of the form $T(C, \mathbf{x}) = A(C, \mathbf{x}) + g(f(A(C, \mathbf{x}); \boldsymbol{\theta}_f); \boldsymbol{\theta}_g)$, a perfect implicit weight update exists for context $Y$ if the inner function $f$ is input-controllable and the outer function $g$ is output-controllable, provided that the activations are non-zero.*

*Proof.* Let $\mathbf{v} = A(C \setminus Y, \mathbf{x})$ and the contextual difference be $\Delta\mathbf{v} = A(C, \mathbf{x}) - A(C \setminus Y, \mathbf{x})$, so $A(C, \mathbf{x}) = \mathbf{v} + \Delta\mathbf{v}$. The original transformer block output is $T(C, \mathbf{x}) = (\mathbf{v} + \Delta\mathbf{v}) + g(f(\mathbf{v} + \Delta\mathbf{v}; \boldsymbol{\theta}_f); \boldsymbol{\theta}_g)$. We seek updated parameters $\boldsymbol{\theta}'_f, \boldsymbol{\theta}'_g$ such that the output with reduced context is identical: $T'(C \setminus Y, \mathbf{x}) = \mathbf{v} + g(f(\mathbf{v}; \boldsymbol{\theta}'_f); \boldsymbol{\theta}'_g)$. The proof proceeds in two steps:

1. **Correct the Input Change**: The input to $f$ changes from $\mathbf{v}$ to $\mathbf{v} + \Delta\mathbf{v}$. Since $f$ is input-controllable, we can find an update $\Delta\boldsymbol{\theta}_f$ such that $f(\mathbf{v} + \Delta\mathbf{v}; \boldsymbol{\theta}_f) = f(\mathbf{v}; \boldsymbol{\theta}_f + \Delta\boldsymbol{\theta}_f)$. Let this common intermediate vector be $\mathbf{z}_{mlp}$.

2. **Correct the Residual Change**: After the first step, the equality we must satisfy is: $(\mathbf{v} + \Delta\mathbf{v}) + g(\mathbf{z}_{mlp}; \boldsymbol{\theta}_g) = \mathbf{v} + g(\mathbf{z}_{mlp}; \boldsymbol{\theta}'_g)$. This simplifies to $g(\mathbf{z}_{mlp}; \boldsymbol{\theta}'_g) - g(\mathbf{z}_{mlp}; \boldsymbol{\theta}_g) = \Delta\mathbf{v}$. This is precisely the definition of output controllability for $g$. Since $g$ is output-controllable, an update $\Delta\boldsymbol{\theta}_g$ exists to satisfy this condition.

With updates to both $\boldsymbol{\theta}_f$ and $\boldsymbol{\theta}_g$, a perfect match is achieved. $\square$

Having established Theorem 3, we can apply it to any new architecture, provided it satisfies the structure described above. We prove input controllability for weight matrix multiplications (of both the direct input and of norms) and output controllability of outer bias, weight matrix multiplication, elementwise multiplication and mixture of experts. We also show an update for parallel transformer blocks. These are all summarized in Table 1 and encompass most common architectures such as Gemma (Kamath et al., 2025), Llama (Touvron et al., 2023), Falcon (Almazrouei et al., 2023), Mistral/Mixtral (Jiang et al., 2023; 2024), Qwen (Yang et al., 2025), GPT-2 (Radford et al., 2019) and GPT-J (Wang and Komatsuzaki, 2021).

## 5.1. Limits of controllability

While our framework accommodates all major modern architectures, its constraints are non-trivial. For instance, if a block uses RMS post-norm *without* a trainable scaling vector $\mathbf{m}$, output controllability fails. Without $\mathbf{m}$, the output is strictly constrained to the $L_2$ sphere, preventing the model from stretching the vector to absorb a contextual shift $\Delta$ whose scale differs from the norm. We provide a formal proof of this impossibility in Appendix D.

## 5.2. Connection to Implicit Gradient Updates

These implicit weight shifts are not merely algebraic rearrangements; they connect directly to standard learning dynamics. Building on Dherin et al. (2025), our context-equivalent updates can be formulated exactly as gradient descent steps on a complex trace-loss objective. In Appendix E we derive this explicitly for both the standard outer-weight-matrix architecture (recovering the Llama/Falcon update via telescoping) and the numerically stable Gemma variant involving the RMSNorm inversion and the scale vector $\mathbf{m}$.

# 6. Conclusion

In this work, we have generalized the theory of implicit weight updates from single layer vanilla transformers to the complex, multi-layer architectures of modern LLMs like Llama (Touvron et al., 2023) and Gemma (Kamath et al., 2025). We began by providing a constructive proof for a single Gemma-style transformer block, deriving the exact rank-1 updates needed to compile context into its MLP weights. We then extended this result to full L-layer models and presented a practical algorithm for computing the updates. We showed practical experiments on Gemma 3 1B and Falcon 7B which achieved almost identical output distributions and the same textual generation.

Finally, we abstracted these findings into the unifying concepts of *input controllability* and *output controllability*. This framework simplifies the analysis and provides a clear theoretical foundation for how modern language model architectures implicitly fine-tune themselves on their context. This work offers a robust and intuitive tool for understanding the mechanisms of in-context learning and for designing future transformer architectures.

We note that our framework provides a descriptive lens for understanding the per-token effect of context, rather than a prescriptive algorithm for efficient inference. The derived updates are token-dependent and must be recomputed at each step to maintain mathematical equivalence. The derived parameter changes do not immediately yield a global update that absorbs the context for every query. This reinforces the view that in-context learning is a dynamic process where the model effectively reconfigures its functional form

for each successive prediction.

## 6.1. Limitations

While our framework establishes a rigorous mathematical equivalence between context and weight updates, its scope has clear boundaries:

- **Token-dependent equivalence:** Our exact updates perfectly replicate the next-token distribution for a specific history. Naively applying these token-specific updates to multi-token free generation is disrupted by the attention values of newly generated tokens.
- **Mechanistic vs. Algorithmic insights:** Reparameterizing context as weight differences allows us to inspect how semantic content is absorbed into weights. However, proving that context *can* be compressed this way does not imply that the transformer explicitly executes this exact learning algorithm during standard inference.
- **Architectural prerequisites:** Our framework requires output controllability of the outer block function. We prove in Appendix D that this fails for post-RMSNorm blocks lacking a trainable scale vector, however this does not appear in any real-world architectures.

## 6.2. Future Work & Applications

Theorem 3 provides the foundation for an automated "context compiler." By traversing the computational graph of any architecture, one could automatically track values and apply mathematically valid controllability updates optimized for specific constraints (e.g., minimizing numerical drift or our desiderata above).

Furthermore, extending our single-token equivalence to unconstrained free generation requires aggregating these transient updates into reusable "thought patches." In concurrent work, Mazzawi et al. (2025) explores this aggregation, directly leveraging the modern architectural updates and numerical stability improvements derived here to successfully absorb complex prompts for multi-token generation.

# Acknowledgments

We would like to thank Hanna Mazzawi, Michael Wunder, Mor Geva, Peter Bartlett and Spencer Frei for their feedback and input into this work.

# Impact Statement

This paper presents work whose goal is to advance the field of Machine Learning, specifically the theoretical understanding and mechanistic interpretability of Large Language Models. Because our primary contribution is foundational and analytical, this work does not present any direct or im-

mediate negative societal consequences. In the long term, we hope our mathematical framework will facilitate the development of more transparent, efficient, and robust AI architectures.

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

# A. General Proofs

## A.1. Proofs of Controllability

**Lemma 1** (Input Controllability of MLPs). *Any function $f(\mathbf{z}; \{W_i\})$ whose initial operations consist of one or more linear projections of the input, such as $f(\ldots, W_i\mathbf{z}, \ldots)$, is input-controllable, provided $\mathbf{z} \neq \mathbf{0}$. This includes standard and gated MLPs (e.g., Gemma's GeGLU, Llama's SwiGLU).*

*Proof.* Let the input change from $\mathbf{z}$ to $\mathbf{z} + \Delta\mathbf{z}$. The arguments to $f$ change from $\{\ldots, W_i\mathbf{z}, \ldots\}$ to $\{\ldots, W_i(\mathbf{z} + \Delta\mathbf{z}), \ldots\}$. We need to find updates $\Delta W_i$ such that the new arguments, using the original input $\mathbf{z}$, are identical: $(W_i + \Delta W_i)\mathbf{z} = W_i(\mathbf{z} + \Delta\mathbf{z})$. This simplifies to $\Delta W_i\mathbf{z} = W_i\Delta\mathbf{z}$. Since $\mathbf{z} \neq \mathbf{0}$, we have $\|\mathbf{z}\|^2 > 0$, so a valid rank-1 update for each matrix $W_i$ is given by:

$$\Delta W_i = \frac{(W_i\Delta\mathbf{z})\mathbf{z}^\top}{\|\mathbf{z}\|^2}$$

Substituting this back gives $\left(\frac{(W_i\Delta\mathbf{z})\mathbf{z}^\top}{\|\mathbf{z}\|^2}\right)\mathbf{z} = W_i\Delta\mathbf{z}\left(\frac{\mathbf{z}^\top\mathbf{z}}{\|\mathbf{z}\|^2}\right) = W_i\Delta\mathbf{z}$. Since updates exist for all input weight matrices, the function is input-controllable. $\square$

**Lemma 2** (Input Controllability of Pre-Norm MLPs). *An MLP function preceded by a normalization layer, $f(N(\mathbf{v}); \{W_i\})$, is input-controllable for changes in the pre-normalized vector $\mathbf{v}$, provided $N(\mathbf{v}) \neq \mathbf{0}$.*

*Proof.* Let the pre-normalized input change from $\mathbf{v}$ to $\mathbf{v}_C$. The normalized input to the MLP changes from $\mathbf{z} = N(\mathbf{v})$ to $\mathbf{z}_C = N(\mathbf{v}_C)$. Let this change be $\Delta\mathbf{z} = \mathbf{z}_C - \mathbf{z}$. The problem then reduces to the case in Lemma 1, where the input to the MLP changes by $\Delta\mathbf{z}$. The required update for each weight matrix $W_i$ is:

$$\Delta W_i = \frac{(W_i(\mathbf{z}_C - \mathbf{z}))\mathbf{z}^\top}{\|\mathbf{z}\|^2}$$

This makes the new input projection $(W_i + \Delta W_i)\mathbf{z} = W_i\mathbf{z}_C$, matching the argument of the function with the original parameters and context-full input. Thus, the pre-norm function is input-controllable. $\square$

**Lemma 3** (Output Controllability of Outer Bias). *The function $g(\mathbf{v}; \mathbf{b}') = h(\mathbf{v}) + \mathbf{b}'$ is output-controllable.*

*Proof.* For a desired output change $\boldsymbol{\delta}$, we need $(h(\mathbf{v}) + \mathbf{b}' + \Delta\mathbf{b}') - (h(\mathbf{v}) + \mathbf{b}') = \boldsymbol{\delta}$. This simplifies to $\Delta\mathbf{b}' = \boldsymbol{\delta}$, which always has a solution. $\square$

**Lemma 4** (Output Controllability of Outer Weight Matrix). *The function $g(\mathbf{v}; W') = W'\mathbf{v}$ is output-controllable when $\mathbf{v} \neq \mathbf{0}$.*

*Proof.* We need $(W' + \Delta W')\mathbf{v} - W'\mathbf{v} = \boldsymbol{\delta}$, which requires $\Delta W'\mathbf{v} = \boldsymbol{\delta}$. A rank-1 update of the form $\Delta W' = \frac{\boldsymbol{\delta}\mathbf{v}^\top}{\|\mathbf{v}\|^2}$ satisfies this condition. $\square$

**Lemma 5** (Output Controllability of Outer Element-wise Multiply). *The function $g(\mathbf{v}; \mathbf{m}) = \mathbf{m} \odot \mathbf{v}$ is output-controllable if no element of $\mathbf{v}$ is zero.*

*Proof.* We need $(\mathbf{m} + \Delta\mathbf{m}) \odot \mathbf{v} - \mathbf{m} \odot \mathbf{v} = \boldsymbol{\delta}$, which implies $\Delta\mathbf{m} \odot \mathbf{v} = \boldsymbol{\delta}$. If no element of $\mathbf{v}$ is zero, this can be solved with element-wise division: $\Delta\mathbf{m} = \boldsymbol{\delta} \oslash \mathbf{v}$. $\square$

**Lemma 6** (Output Controllability of Mixture of Experts). *A Mixture of Experts (MoE) layer of the form $g(\mathbf{v}; \{\boldsymbol{\theta}_j\}, \mathbf{s}) = \sum_{j=1}^{N} s_j \cdot Ex_j(\mathbf{v}; \boldsymbol{\theta}_j)$, where $s_j$ are router gates and $Ex_j$ are expert networks, is output-controllable if each expert $Ex_j$ is output-controllable and the sum of gate values $S = \sum s_j \neq 0$.*

*Proof.* Let the desired output change be $\boldsymbol{\delta}$. We distribute this change across the experts, setting the target change for each expert $j$ to be $\boldsymbol{\delta}_j = \boldsymbol{\delta}/S$. Since each expert $Ex_j$ is output-controllable by assumption, a parameter update $\Delta\boldsymbol{\theta}_j$ exists to produce this change. The total change in the MoE output is then $\sum s_j \cdot (\boldsymbol{\delta}/S) = (\sum s_j) \cdot (\boldsymbol{\delta}/S) = S \cdot (\boldsymbol{\delta}/S) = \boldsymbol{\delta}$. $\square$

**Lemma 7** (Implicit Updates in Parallel Blocks). *For a parallel transformer block of the form $T(C, \mathbf{x}) = \mathbf{x} + A(C, \mathbf{x}) + g(f(\mathbf{x}); \boldsymbol{\theta}_g)$, a perfect implicit weight update exists if the outer function $g$ is output-controllable.*

*Proof.* In this architecture, the MLP branch $g(f(\mathbf{x}))$ is context-independent. Noting that the input $\mathbf{x}$ cancels from both sides of the block equation, the entire contextual difference from the parallel attention branch, $\Delta A_{\mathbf{x}}(Y)$, must be absorbed by the MLP branch. For the outputs to be equal, we require: $A(C, \mathbf{x}) + g(f(\mathbf{x}); \boldsymbol{\theta}_g) = A(C \setminus Y, \mathbf{x}) + g(f(\mathbf{x}); \boldsymbol{\theta}_g')$. This simplifies to $g(f(\mathbf{x}); \boldsymbol{\theta}_g') - g(f(\mathbf{x}); \boldsymbol{\theta}_g) = \Delta A_{\mathbf{x}}(Y)$. This is the definition of output controllability for $g$. $\qquad\square$

# B. Appendix: Numerically Stable Update and RMSNorm Inversion

## B.1. Numerically Stable Update via RMSNorm Inversion

The direct update for $\Delta \mathbf{m}$ derived in Theorem 1 can be numerically unstable, as it involves element-wise division by the MLP's output, which may contain values at or near zero. The "Stable" update referenced in Section 4 mitigates this by primarily updating the $W_{\text{down}}$ matrix, using $\Delta \mathbf{m}$ only to absorb any minor remaining error.

This method works by inverting the final $\mathbf{m} \odot \text{Norm}(\cdot)$ operation. Let the inputs to the $W_{\text{down}}$ layer (after the $W_{\text{gate/up}}$ updates are applied) be $\mathbf{h}_{\text{gated},C}$. Let the original pre-normalization vector be $\mathbf{h}_{\text{down},C} = W_{\text{down}} \mathbf{h}_{\text{gated},C}$, and the original scaled, normalized output be $\mathbf{h}_{\text{out},C} = \mathbf{m} \odot \text{Norm}(\mathbf{h}_{\text{down},C})$.

The goal is to find updates $\Delta W_{\text{down}}$ and $\Delta \mathbf{m}$ such that the new output component matches the target $\mathbf{g} = (\mathbf{v}_C - \mathbf{v}) + \mathbf{h}_{\text{out},C}$.

The process is as follows:

1. **Find Target Pre-Norm Vector.** We first find an optimal pre-normalization vector $\mathbf{h}_{\text{target}}$ that, when normalized and scaled, best approximates $\mathbf{g}$. This is achieved using the analytical RMSNorm inversion derived in Appendix B.2. We set the target RMS to the original RMS value, $C = \text{RMS}(\mathbf{h}_{\text{down},C})$.

$$\mathbf{h}_{\text{target}} = \text{InvertRMSNorm}(\mathbf{g}, \mathbf{m}, C)$$

   This function finds the $\mathbf{h}_{\text{target}}$ that minimizes $\|\mathbf{m} \odot \text{Norm}(\mathbf{h}_{\text{target}}) - \mathbf{g}\|^2$ under the constraint $\text{RMS}(\mathbf{h}_{\text{target}}) = C$. [1]

2. **Update $W_{\text{down}}$.** We compute a rank-1 update $\Delta W_{\text{down}}$ to absorb the difference $\boldsymbol{\delta} = \mathbf{h}_{\text{target}} - \mathbf{h}_{\text{down},C}$, ensuring the $W_{\text{down}}$ layer now outputs $\mathbf{h}_{\text{target}}$.

$$\Delta W_{\text{down}} = \frac{\boldsymbol{\delta} \cdot \mathbf{h}_{\text{gated},C}^{\top}}{\|\mathbf{h}_{\text{gated},C}\|^2}$$

3. **Calculate Remainder and Update $\mathbf{m}$.** The inversion in Step 1 is an L2-minimizing approximation, not necessarily an exact match. Let the new pre-norm vector be $\mathbf{h}_{\text{down}}' = (W_{\text{down}} + \Delta W_{\text{down}}) \mathbf{h}_{\text{gated},C} = \mathbf{h}_{\text{target}}$. Let its normalized form be $\mathbf{h}_{\text{norm}}' = \text{Norm}(\mathbf{h}_{\text{down}}')$.
   The remaining error is $\mathbf{r} = \mathbf{g} - (\mathbf{m} \odot \mathbf{h}_{\text{norm}}')$. We absorb this small remainder with $\Delta \mathbf{m}$:

$$(\mathbf{m} + \Delta \mathbf{m}) \odot \mathbf{h}_{\text{norm}}' = \mathbf{g} \implies \Delta \mathbf{m} = \mathbf{r} \oslash \mathbf{h}_{\text{norm}}'$$

   This final division is more stable because the numerator $\mathbf{r}$ is expected to be very small, counteracting the impact of small values in $\mathbf{h}_{\text{norm}}'$.

## B.2. Derivation of Analytical RMSNorm Inversion

We seek to find a vector $\mathbf{x} \in \mathbb{R}^n$ that minimizes the squared $L_2$ error to a target vector $\mathbf{g}$, after applying scaled RMS normalization, subject to a fixed RMS value.

**Definition 3** (Scaled RMSNorm). *The scaled RMSNorm function is defined as:*

$$RMSNorm(\mathbf{x}, \mathbf{m}) = \left(\frac{\mathbf{x}}{RMS(\mathbf{x})}\right) \odot \mathbf{m}$$

*where $RMS(\mathbf{x}) = \sqrt{\frac{1}{n} \sum_{i=1}^{n} x_i^2} = \frac{\|\mathbf{x}\|}{\sqrt{n}}$.*

---

[1]We note that other choices for $\mathbf{h}_{\text{target}}$ are also possible such as $\mathbf{h}_{\text{target}} = \alpha \cdot \mathbf{g} \oslash \mathbf{m}$ scaled to $\text{RMS}(\mathbf{h}_{\text{target}}) = C$, this variant is shown in Appendix C.

### B.2.1. PROBLEM FORMULATION

The objective is to find $\mathbf{x}$ that minimizes

$$L(\mathbf{x}) = \left\| \left( \frac{\mathbf{x}}{\text{RMS}(\mathbf{x})} \right) \odot \mathbf{m} - \mathbf{g} \right\|^2$$

subject to the constraint $\text{RMS}(\mathbf{x}) = C$ for a known constant $C > 0$. This constraint reduces from an infinite set of solutions to a single one.

To simplify, let $\mathbf{y} = \mathbf{x}/C$. Then $\text{RMS}(\mathbf{y}) = 1$, and $\mathbf{x} = C\mathbf{y}$. Substituting this into the objective yields an equivalent problem: find the vector $\mathbf{y}$ that minimizes

$$f(\mathbf{y}) = \|\mathbf{y} \odot \mathbf{m} - \mathbf{g}\|^2 = \sum_{k=1}^{n} (y_k m_k - g_k)^2 \tag{5}$$

subject to the constraint

$$h(\mathbf{y}) = \text{RMS}(\mathbf{y})^2 - 1 = \frac{1}{n} \sum_{k=1}^{n} y_k^2 - 1 = 0 \tag{6}$$

### B.2.2. SOLUTION VIA LAGRANGE MULTIPLIERS

We solve this constrained optimization problem using the method of Lagrange multipliers. The Lagrangian function $\mathcal{L}(\mathbf{y}, \lambda)$ is:

$$\mathcal{L}(\mathbf{y}, \lambda) = f(\mathbf{y}) - \lambda h(\mathbf{y})$$
$$= \sum_{k=1}^{n} (y_k m_k - g_k)^2 - \lambda \left( \frac{1}{n} \sum_{k=1}^{n} y_k^2 - 1 \right)$$

To find the optimal $\mathbf{y}$, we set the gradient of $\mathcal{L}$ with respect to each component $y_k$ to zero:

$$\frac{\partial \mathcal{L}}{\partial y_k} = \frac{\partial}{\partial y_k} (y_k m_k - g_k)^2 - \lambda \frac{\partial}{\partial y_k} \left( \frac{1}{n} y_k^2 \right) = 0$$

Using the chain rule:

$$2(y_k m_k - g_k) \cdot m_k - \lambda \left( \frac{2 y_k}{n} \right) = 0$$

Dividing by 2 and rearranging to solve for $y_k$:

$$y_k m_k^2 - g_k m_k = \frac{\lambda}{n} y_k$$

$$y_k \left( m_k^2 - \frac{\lambda}{n} \right) = g_k m_k$$

Letting $\mu = \lambda/n$, we find the form of the solution for each component:

$$y_k = \frac{g_k m_k}{m_k^2 - \mu} \tag{7}$$

The scalar $\mu$ is a constant related to the Lagrange multiplier, which must be chosen to satisfy the constraint $h(\mathbf{y}) = 0$.

### B.2.3. FINDING THE MULTIPLIER $\mu$

We find $\mu$ by substituting the solution form from Equation (7) back into the constraint Equation (6):

$$\frac{1}{n} \sum_{k=1}^{n} \left( \frac{g_k m_k}{m_k^2 - \mu} \right)^2 = 1$$

Thus, $\mu$ must be the root of the function $F(\mu) = 0$, where:

$$F(\mu) = \left( \frac{1}{n} \sum_{k=1}^{n} \frac{(g_k m_k)^2}{(m_k^2 - \mu)^2} \right) - 1 \tag{8}$$

To guarantee a unique solution, we analyze $F(\mu)$ on the interval $\mathcal{I} = (-\infty, \min_k(m_k^2))$. This interval ensures the denominator $(m_k^2 - \mu)$ is always positive and non-zero.

**1. Existence of a Root.** We check the limits of $F(\mu)$ at the boundaries of $\mathcal{I}$:

- As $\mu \to -\infty$, the denominator $(m_k^2 - \mu)^2 \to +\infty$ for all $k$, so each term in the sum approaches 0. Thus, $\lim_{\mu \to -\infty} F(\mu) = -1$.

- As $\mu \to (\min_k m_k^2)^-$, at least one denominator term $(m_k^2 - \mu)^2 \to 0^+$, causing the sum to diverge. Thus, $\lim_{\mu \to (\min m_k^2)^-} F(\mu) = +\infty$.

Since $F(\mu)$ is continuous on $\mathcal{I}$ and transitions from a negative to a positive value, the Intermediate Value Theorem guarantees that at least one root exists in this interval.

**2. Uniqueness of the Root.** We show the root is unique by proving $F(\mu)$ is strictly monotonic on $\mathcal{I}$. We analyze its derivative, $F'(\mu)$:

$$\begin{aligned}
F'(\mu) &= \frac{d}{d\mu} \left[ \left( \frac{1}{n} \sum_{k=1}^{n} (g_k m_k)^2 (m_k^2 - \mu)^{-2} \right) - 1 \right] \\
&= \frac{1}{n} \sum_{k=1}^{n} (g_k m_k)^2 \cdot \left( -2(m_k^2 - \mu)^{-3} \cdot (-1) \right) \\
&= \frac{2}{n} \sum_{k=1}^{n} \frac{(g_k m_k)^2}{(m_k^2 - \mu)^3}
\end{aligned}$$

On the interval $\mathcal{I}$, we have $m_k^2 - \mu > 0$ for all $k$. Therefore, $(g_k m_k)^2 \geq 0$ and $(m_k^2 - \mu)^3 > 0$. Assuming a non-trivial case where not all $g_k m_k = 0$, the derivative $F'(\mu)$ is a sum of positive terms, so $F'(\mu) > 0$.

Since $F(\mu)$ is strictly monotonically increasing on $\mathcal{I}$, it can cross the axis only once. Thus, a unique root $\mu$ exists and can be found efficiently using a numerical method such as bisection search.

Once $\mu$ is found, the optimal normalized vector $\mathbf{y}$ is given by Equation (7), and the final unnormalized vector $\mathbf{x}$ is recovered by scaling by the goal RMS norm $C$: $\mathbf{x} = C \cdot \mathbf{y}$.

## C. Extended Experimental Results

In this section, we provide a comprehensive breakdown of the experimental validation across different prompts, metric categories, and ablation studies regarding layer selection and scaling updates.

### C.1. Additional Prompts

To ensure the robustness of our findings beyond the primary "Mars weather" example, we evaluated the update mechanism on four additional distinct prompts ranging from creative writing to analytical tasks. Figure 6 and Figure 7 show the generation metrics for these prompts. In all cases using `float32`, we observe near-zero Logit Difference and Total Variation Distance.

### C.2. Extended Metrics Analysis

Beyond the standard accuracy and TVD reported in the main text, we analyze the structural impact of the updates on the model parameters and output distributions.

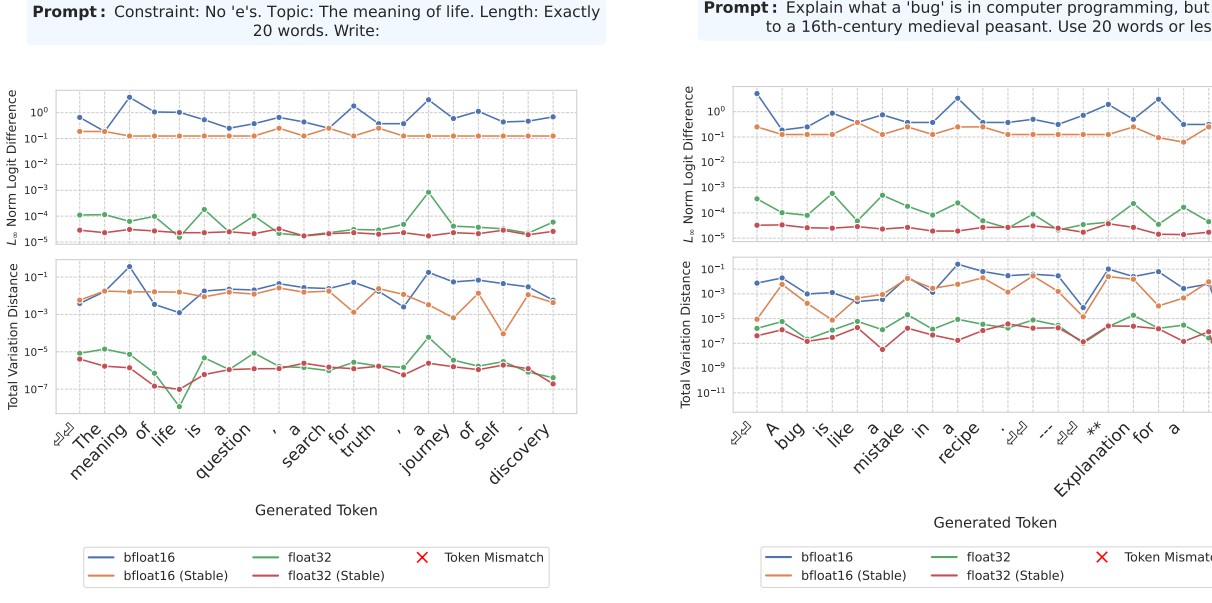

*(a)* Prompt 1: Constrained generation (note that the original model also ignores the constraint).

*(b)* Prompt 2: Analogy creation

*Figure 6.* Generation metrics for Prompts 1 and 2. The updated model (no context) maintains high fidelity to the original model (with context) across different textual domains.

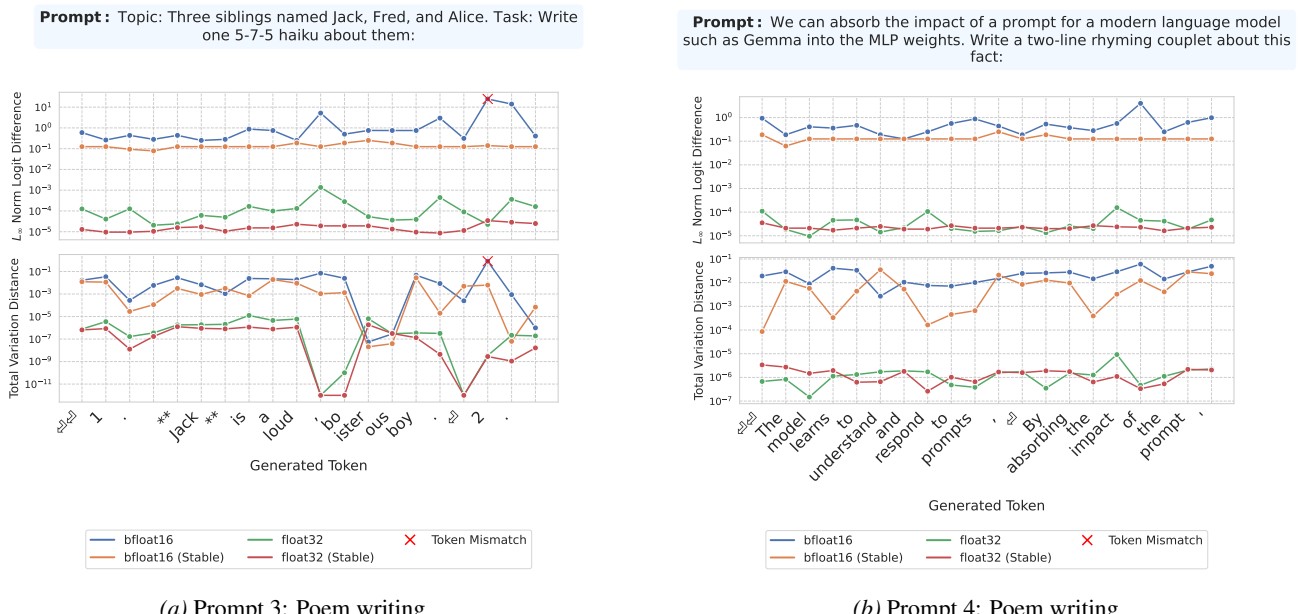

*(a)* Prompt 3: Poem writing

*(b)* Prompt 4: Poem writing

*Figure 7.* Generation metrics for Prompts 3 and 4. The updated model (no context) maintains high fidelity to the original model (with context) across different textual domains.

Figure 8 displays two distinct views. Figure 8a illustrates distance metrics including Hellinger distance and Rank distance, confirming that the probability landscapes remain aligned. Figure 8b tracks the Frobenius norms of the weight updates and the L2 norms of the vector updates across layers, showing that the required patches are generally sparse and low-magnitude. We define these metrics below.

- **Hellinger distance:** The Hellinger distance is defined as $H(p,q) = \frac{1}{\sqrt{2}}\|\sqrt{p} - \sqrt{q}\|_2$

- **Rank distance:** The rank distance is the Pearson correlation between logit orderings $\rho(p, q) = \frac{\text{cov}(R(p), R(q))}{\sigma_{R(p)} \sigma_{R(q)}}$, where $R$ is the rank vector $R(v)_i = |\{j : (v_j, j) < (v_i, i)\}|$ and $\sigma$ is the standard deviation.
- **Top-1 difference:** The top-1 difference is the difference between the probability of the next predicted token (according to the original model) under the two settings. Top-1-Diff$(p, q) = \max p - q_{\text{argmax } p}$.
- **Matrix update norm:** The Frobenius norm of weight matrix updates across all layers
- **Vector update norm:** The L2 norm of vector updates (scale vector in RMSNorm) across all layers

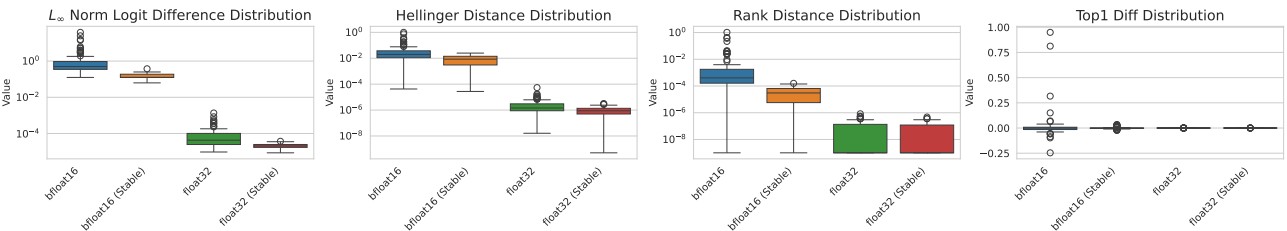

*(a)* Distance Metrics (Hellinger, Rank Correlation)

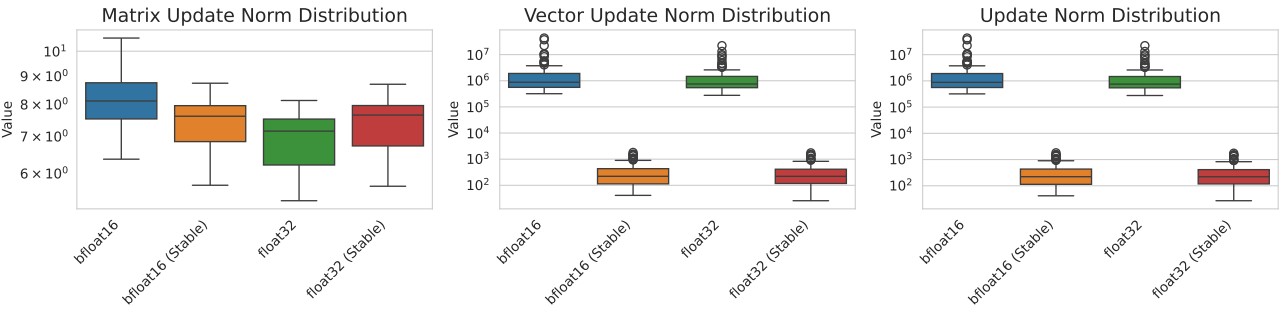

*(b)* Update Norms (Matrix Frobenius, Vector L2)

*Figure 8.* Detailed analysis of generation divergence and parameter update magnitudes. The low distance metrics confirm distribution matching, while the norm plots indicate that the minimization of the norm in Appendix B results in a much smaller update

## C.3. Empirical Validation on Falcon

We replicated the evaluation suite on the Falcon-7B architecture to verify the model-agnostic nature of our controllability framework.

**Architectural Note.** Falcon utilizes a pre-norm architecture without post-normalization scaling, which simplifies the controllability update procedure compared to Gemma-style blocks. Specifically, the update does not require the approximate RMSNorm inversion derived in Appendix B. Consequently, the implicit updates on Falcon are inherently more numerically stable, as they avoid the "exploding" updates seen in Gemma when operating in low-precision formats.

**Experimental Results.** Figure 9 presents the aggregated metrics for Falcon across all prompts, comparing `float32` and `bfloat16` precision. Unlike the Gemma results, Falcon achieves perfect token-matching even in base `bfloat16` without requiring stabilization techniques. The updated models consistently achieve token-level fidelity and negligible logit drift.

## C.4. Ablation: Starting Layer Depth

As noted in Section 3.2, it is theoretically possible to absorb context using only a subset of the final layers. Figure 11 and Figure 12 investigate the numerical stability of this approach.

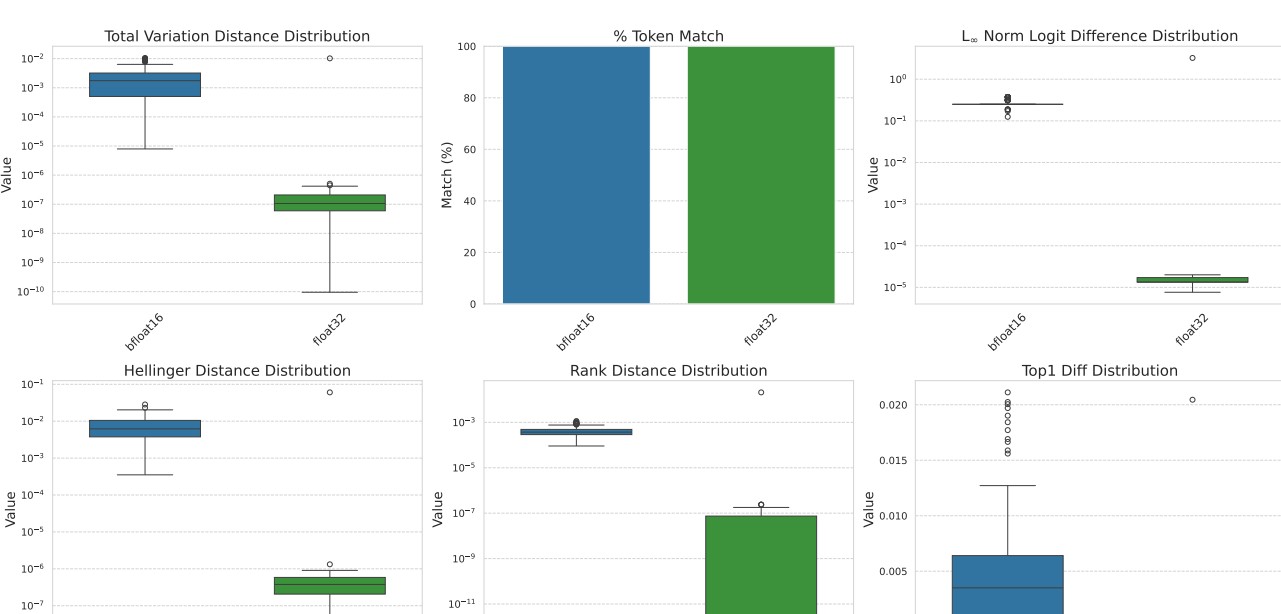

*Figure 9.* Falcon-7B performance metrics aggregated across all prompts and precision settings. The combined grid displays distributions for TVD, Token Match percentage, $L_\infty$ Logit difference, Hellinger distance, Rank distance, and Top-1 difference.

We observe that updating only the final layer presents numerical issues. However, provided there are a few layers available to make adjustments, the impact on accuracy is minimal. Figure 12 compares the naive update against the numerically stable update (via RMSNorm inversion) when varying the starting layer. We find that updating the last layer is consistently an issue. We also investigate the norms of the updates, finding that while the matrix norm update decreases as we update fewer layers, the vector norm increases (under the stable regime) and is multiple orders of magnitude larger. As such the smallest norm update results from starting at earlier layers.

### C.5. Analysis of Scaling Update Variant

Finally, we evaluate the robustness of the different scaling update strategies discussed in Appendix B. We compare the standard element-wise division and numerically stable update against a version which updates $W_{\text{down}}$ but only scales the RMSNorm parameter. Other choices for $\mathbf{h}_{\text{target}}$ are also possible such as $\mathbf{h}_{\text{target}} = \alpha \cdot \mathbf{g} \oslash \mathbf{m}$ scaled to $\text{RMS}(\mathbf{h}_{\text{target}}) = C$.

Figure 13 presents a three-part view:

- **(a) Main Metrics:** Tracks $L_\infty$ norm, TVD, and token matching accuracy across `bfloat16` and `float32`.

- **(b) Norm Analysis:** Displays the magnitude of the resulting $\Delta\mathbf{m}$ and $\Delta W$ vectors.

- **(c) Auxiliary Metrics:** Shows additional divergence measures.

The results demonstrate that the scaled update variant minimizes extreme values in $\Delta\mathbf{m}$, leading to better preservation of the token distribution in lower precision.

## D. Impossibility of Controllability for RMS Post-Norm Without a Trainable Scaling Vector

In this section, we demonstrate that output controllability (Definition 2) requires specific architectural flexibilities. While Theorem 3 establishes that an exact implicit update exists for most modern models, this relies on the outer function $g$ being output-controllable. If an architecture lacks a trainable parameter at its output boundary, this property can fail.

**Prompt:** Write a single-sentence weather forecast for Mars, from the perspective of a slightly annoyed robot:

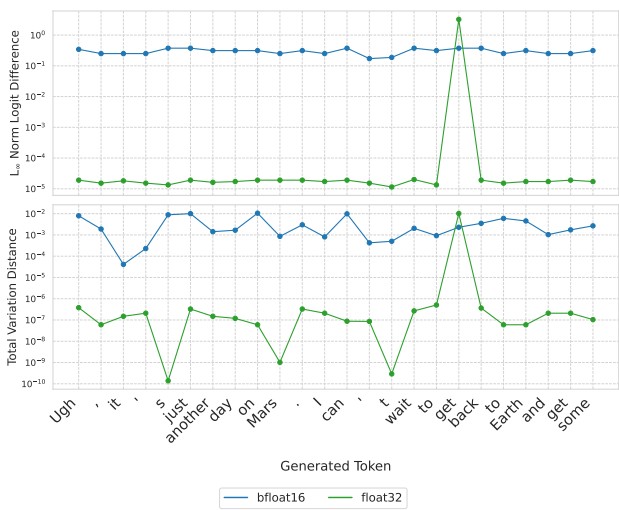

*(a)* Mars Weather Forecast (Annoyed Robot)

**Prompt:** Constraint: No 'e's. Topic: The meaning of life. Length: Exactly 20 words. Write:

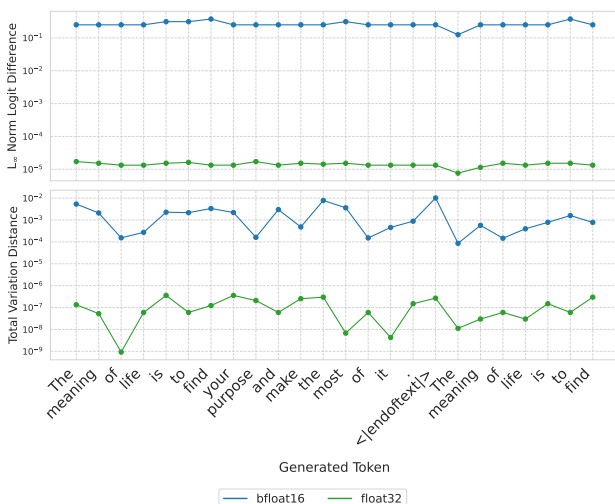

*(b)* Meaning of Life (Constraint: No 'e')

**Prompt:** Explain what a 'bug' is in computer programming, but explain it to a 16th-century medieval peasant. Use 20 words or less:

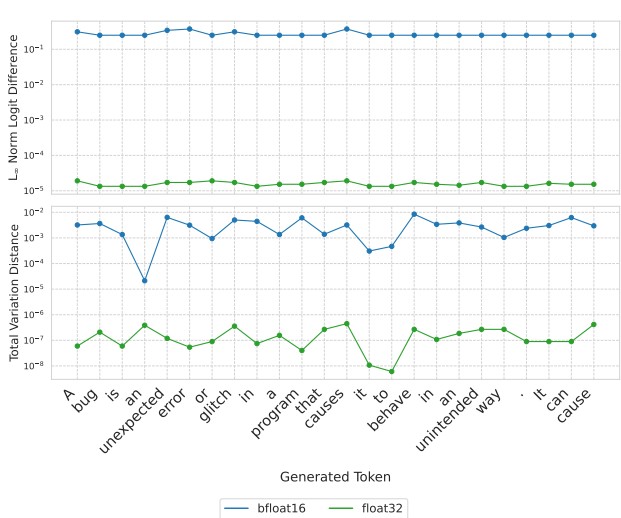

*(c)* Explain 'Bug' (Medieval Peasant)

**Prompt:** Topic: Three siblings named Jack, Fred, and Alice. Task: Write one 5-7-5 haiku about them

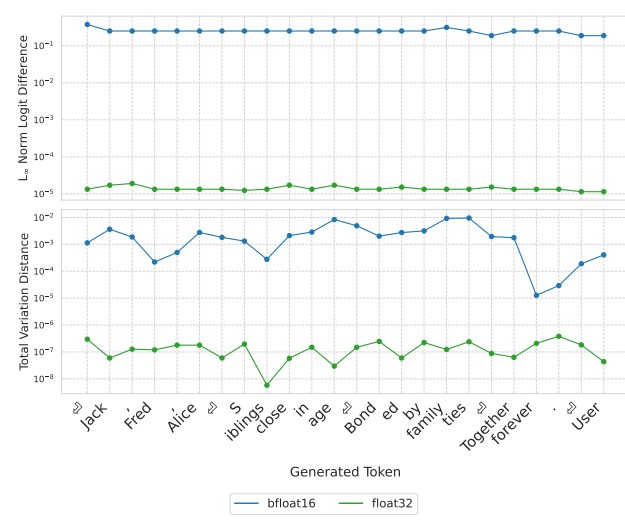

*(d)* Sibling Haiku

**Prompt:** We can absorb the impact of a prompt for a modern language model such as Gemma into the MLP weights. Write a two-line rhyming couplet about this fact:

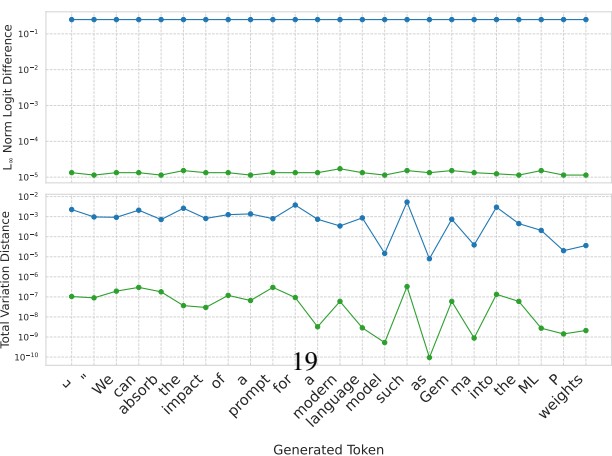

**Theorem 4** (Impossibility of Fixed Post-Normalization). *For a post-normalization residual block of the form $T(C, \mathbf{x}) = A(C, \mathbf{x}) + g(f(A(C, \mathbf{x})))$, an exact context-equivalent update cannot be guaranteed to exist if the outer function $g$ is an RMSNorm operation with a fixed (non-trainable) scaling vector $\mathbf{m}$.*

*Proof.* Let the original transformer block be evaluated with full context $C$, and the updated block be evaluated with the reduced context $C \setminus Y$. Following the logic of Theorem 3, for the final outputs to be perfectly identical, the MLP branch must exactly absorb the contextual difference vector from the attention branch, $\Delta A_{\mathbf{x}}(Y) = A(C, \mathbf{x}) - A(C \setminus Y, \mathbf{x})$.

Mathematically, there must exist an updated internal activation $\mathbf{z}'$ such that:

$$g(\mathbf{z}') - g(\mathbf{z}) = \Delta A_{\mathbf{x}}(Y) \tag{9}$$

where $\mathbf{z}$ is the original internal activation and $g(\mathbf{z}) = \text{RMSNorm}(\mathbf{z})$. Note that $\mathbf{z}'$ may be produced by any update to the parameters of $f$ (and to the preceding $W_{\text{gate}}, W_{\text{up}}$ matrices); the bound on $\|g(\mathbf{z}') - g(\mathbf{z})\|_2$ derived below holds uniformly over the choice of $\mathbf{z}'$, so no update upstream of $g$ can rescue the equality. By definition, the RMSNorm of a vector $\mathbf{z} \in \mathbb{R}^d$ with a fixed scale vector $\mathbf{m}$ is:

$$g(\mathbf{z}) = \mathbf{m} \odot \frac{\mathbf{z}}{\text{RMS}(\mathbf{z})} = \mathbf{m} \odot \left( \sqrt{d} \frac{\mathbf{z}}{\|\mathbf{z}\|_2} \right) \tag{10}$$

Because the normalized vector $\frac{\mathbf{z}}{\|\mathbf{z}\|_2}$ lies on the unit hypersphere, the $L_2$ norm of the output of $g$ is strictly bounded by the maximum element of $\mathbf{m}$ ($\|\mathbf{m}\|_\infty = \max_i |m_i|$):

$$\|g(\mathbf{z})\|_2 \leq \sqrt{d}\|\mathbf{m}\|_\infty \tag{11}$$

By the triangle inequality, the maximum possible change the MLP branch can produce is strictly bounded by the diameter of this output space:

$$\|g(\mathbf{z}') - g(\mathbf{z})\|_2 \leq \|g(\mathbf{z}')\|_2 + \|g(\mathbf{z})\|_2 \leq 2\sqrt{d}\|\mathbf{m}\|_\infty \tag{12}$$

This establishes a firm geometric upper bound on the corrective capacity of the MLP branch. The norm $\|\Delta A_x(Y)\|_2$, on the other hand, has no such architectural bound: it is produced by the attention sub-layer, whose output magnitude is governed by its own (independent) projection and scale parameters. Concretely, scaling the previous block's output projection by $\alpha$ scales $\|A(C, x)\|_2$ and hence $\|\Delta A_x(Y)\|_2$ by a factor of $\alpha$, while leaving the bound $2\sqrt{d}\|\mathbf{m}\|_\infty$ on $g$'s image unchanged. Choosing $\alpha$ large enough yields $\|\Delta A_x(Y)\|_2 > 2\sqrt{d}\|\mathbf{m}\|_\infty$.

Consequently, the outer function $g$ is not output-controllable, and exact single-token equivalence is mathematically impossible unless $\mathbf{m}$ can be modified as a trainable parameter (Lemma 5). $\square$

# E. Deriving the Implicit Update as a Gradient Step

As noted in Section 5, our context-equivalent updates can be viewed as gradient descent steps on a specific loss objective. Building on the trace-loss formulation from Dherin et al. (2025), we define an implicit objective measuring the discrepancy between the uncontextualized state and the contextualized target.

To demonstrate this, we define a step-wise context accumulation from $i = 0$ (no context) to $i = n$ (full context $C$). At each context step $i \to i + 1$, the model's parameters $\boldsymbol{\Theta}$ take a gradient descent step on a global meta-loss $\mathcal{L}_i(\boldsymbol{\Theta})$:

$$\boldsymbol{\Theta}_{i+1} = \boldsymbol{\Theta}_i - H\nabla_{\boldsymbol{\Theta}}\mathcal{L}_i(\boldsymbol{\Theta}_i) \tag{13}$$

where $H$ represents a parameter-specific learning rate scaling.

## E.1. Notation and Setup

For a given layer (dropping the layer index $k$ for readability) and context step $i \in \{0, \ldots, n\}$:

- $\mathbf{v}_i$: The pre-norm residual vector at step $i$. ($\mathbf{v}_0 \equiv \mathbf{v}$ for no context; $\mathbf{v}_n \equiv \mathbf{v}_C$ for full context).
- $\mathbf{z}_i = N_{\text{RMS}}(\mathbf{v}_i)$: The post-norm input vector.
- $W_{\text{gate},0}, W_{\text{up},0}, W_{\text{down},0}$: The frozen, pre-trained initial weights.
- $\mathbf{h}_{\text{mlp},C}$: The target internal MLP activation (after the activation function) calculated using the full context $n$.

To ensure the gradient descent step ($-\nabla\mathcal{L}$) yields our additive updates, we utilize a trace inner product $\langle \Delta, W \rangle = \text{Tr}(\Delta^\top W)$. If we define a pseudo-loss $\mathcal{L}(W) = -\text{Tr}(\Delta^\top W)$, its gradient is precisely $-\Delta$, yielding an update step of $+\Delta$.

**E.2. Case 1: Standard Outer Weight Matrix (e.g., Llama-style)**

For an architecture utilizing a standard outer weight matrix without a trainable RMSNorm scale vector mapping directly back to the residual stream, we define the global meta-loss at step $i$ as:

$$\mathcal{L}_{\text{Llama},i}(\boldsymbol{\Theta}) = -\text{Tr}\big(\Delta_{\text{gate},i}^{\top} W_{\text{gate}}\big) - \text{Tr}\big(\Delta_{\text{up},i}^{\top} W_{\text{up}}\big) - \text{Tr}\big(\Delta_{\text{down},i}^{\top} W_{\text{down}}\big) \tag{14}$$

The target shift matrices ($\Delta$) are defined to absorb the vector differences at each step. For the input matrices (matching Input Controllability, Lemma 2):

$$\Delta_{\text{gate},i} = W_{\text{gate},0}\big(\mathbf{z}_{i+1} - \mathbf{z}_i\big)\mathbf{z}_0^{\top} \tag{15}$$

$$\Delta_{\text{up},i} = W_{\text{up},0}\big(\mathbf{z}_{i+1} - \mathbf{z}_i\big)\mathbf{z}_0^{\top} \tag{16}$$

For the output matrix (matching Output Controllability, Lemma 4), the shift absorbs the difference in the pre-norm residual space:

$$\Delta_{\text{down},i} = \big(\mathbf{v}_{i+1} - \mathbf{v}_i\big)\mathbf{h}_{\text{mlp},C}^{\top} \tag{17}$$

**The Gradient Descent Step.** The learning rates $H$ are scalar values defined by the inverse squared norms of the respective input vectors: $\eta_{\text{in}} = 1/\|\mathbf{z}_0\|^2$ and $\eta_{\text{out}} = 1/\|\mathbf{h}_{\text{mlp},C}\|^2$. The update for the gate matrix is:

$$W_{\text{gate},i+1} = W_{\text{gate},i} - \eta_{\text{in}}\nabla_{W_{\text{gate}}}\mathcal{L}_{\text{Llama},i} = W_{\text{gate},i} + \eta_{\text{in}}\Delta_{\text{gate},i} \tag{18}$$

By summing these gradient steps from $i = 0$ to $n - 1$, the intermediate terms telescope perfectly:

$$\sum_{i=0}^{n-1}(\mathbf{z}_{i+1} - \mathbf{z}_i) = \mathbf{z}_n - \mathbf{z}_0 = \mathbf{z}_C - \mathbf{z} \tag{19}$$

This perfectly yields the cumulative single-step update derived in the main text: $\Delta W_{\text{gate}} = \frac{W_{\text{gate},0}(\mathbf{z}_C - \mathbf{z})\mathbf{z}_0^{\top}}{\|\mathbf{z}_0\|^2}$.

**E.3. Case 2: Numerically Stable Update (e.g., Gemma-style)**

For architectures where we utilize the RMSNorm Inversion derived in Appendix B, $W_{\text{down}}$ targets an optimal pre-norm vector $\mathbf{h}_{\text{target}}$, and the RMSNorm scale vector $\mathbf{m}$ absorbs the remaining error $\mathbf{r}$. Let $\mathbf{h}_{\text{gated},C}$ be the input to the down projection, and $\mathbf{h}'_{\text{norm}}$ be the normalized output of the updated $W_{\text{down}}$ layer. The meta-loss is updated to accommodate the vector derivative for $\mathbf{m}$:

$$\mathcal{L}_{\text{Gemma},i}(\boldsymbol{\Theta}) = -\text{Tr}\big(\Delta_{\text{gate},i}^{\top} W_{\text{gate}}\big) - \text{Tr}\big(\Delta_{\text{up},i}^{\top} W_{\text{up}}\big) - \text{Tr}\big(\Delta_{\text{down},i}^{\top} W_{\text{down}}\big) - \Delta_{\mathbf{m},i}^{\top}\big(\mathbf{m} \odot \mathbf{h}'_{\text{norm}}\big) \tag{20}$$

The input matrices ($\Delta_{\text{gate}}$ and $\Delta_{\text{up}}$) remain identical to the previous case. The output shifts now target the inverted goals:

$$\Delta_{\text{down},i} = \big(\mathbf{h}_{\text{target},i+1} - \mathbf{h}_{\text{target},i}\big)\mathbf{h}_{\text{gated},C}^{\top} \tag{21}$$

$$\Delta_{\mathbf{m},i} = \mathbf{r}_{i+1} - \mathbf{r}_i \tag{22}$$

**The Gradient Descent Step.** The matrices update using scalar learning rates as before (with $\eta_{\text{down}} = 1/\|\mathbf{h}_{\text{gated},C}\|^2$). However, the scale vector $\mathbf{m}$ requires an **element-wise learning rate vector** to invert the Hadamard product:

$$\vec{\eta}_{\mathbf{m}} = \mathbf{1} \oslash \big(\mathbf{h}'_{\text{norm}} \odot \mathbf{h}'_{\text{norm}}\big) \tag{23}$$

Here $h'_{\text{norm}}$ is held fixed at its terminal value $\text{Norm}(h_{\text{target},n})$ across all steps $i$; equivalently, we choose the element-wise learning rate $\vec{\eta}_m = 1 \oslash (h'_{\text{norm}} \odot h'_{\text{norm}})$ as a fixed preconditioner independent of $i$. This is consistent with the algorithm's construction, where the $W_{\text{down}}$ updates by design land on $h_{\text{target},i}$ at each step, decoupling the $m$ subproblem from the running $W_{\text{down}}$ state. Without this choice, the element-wise division would not commute with the summation and the telescoping would fail.

Noting that $\nabla_{\mathbf{m}}[-\Delta_{\mathbf{m}}^\top(\mathbf{m} \odot \mathbf{h}'_{\text{norm}})] = -\Delta_{\mathbf{m}} \odot \mathbf{h}'_{\text{norm}}$, the gradient descent step for $\mathbf{m}$ is an element-wise coordinate descent:

$$\mathbf{m}_{i+1} = \mathbf{m}_i - \vec{\eta}_{\mathbf{m}} \odot \nabla_{\mathbf{m}} \mathcal{L}_{\text{Gemma},i} \tag{24}$$

$$= \mathbf{m}_i - \vec{\eta}_{\mathbf{m}} \odot \left( -\Delta_{\mathbf{m},i} \odot \mathbf{h}'_{\text{norm}} \right) \tag{25}$$

$$= \mathbf{m}_i + \left( \mathbf{r}_{i+1} - \mathbf{r}_i \right) \oslash \mathbf{h}'_{\text{norm}} \tag{26}$$

Summing this from $i = 0$ to $n - 1$ causes the remainders to telescope: $\sum(\mathbf{r}_{i+1} - \mathbf{r}_i) = \mathbf{r}_n - \mathbf{r}_0$. Because the initial remainder error $\mathbf{r}_0 = \mathbf{0}$, this globally collapses into our derived stable update formula: $\Delta\mathbf{m} = \mathbf{r}_n \oslash \mathbf{h}'_{\text{norm}}$.

Summary Statistics Over All Generated Tokens

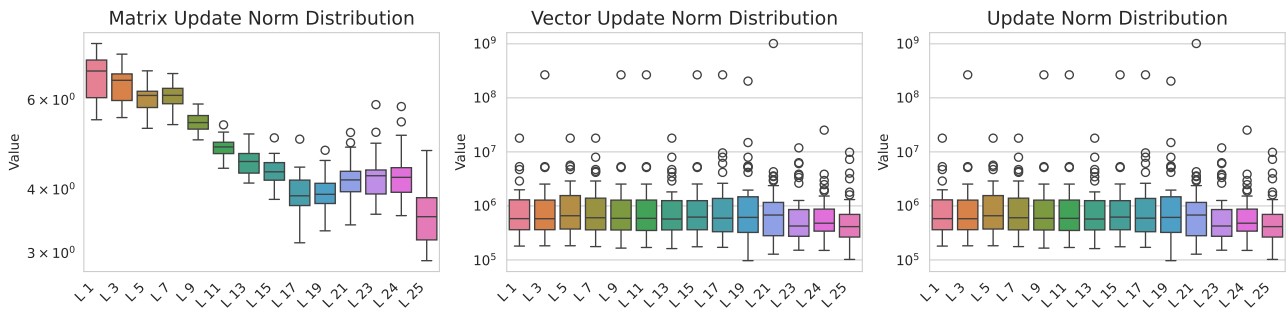

*(a)* `float32` / Naive

Summary Statistics Over All Generated Tokens

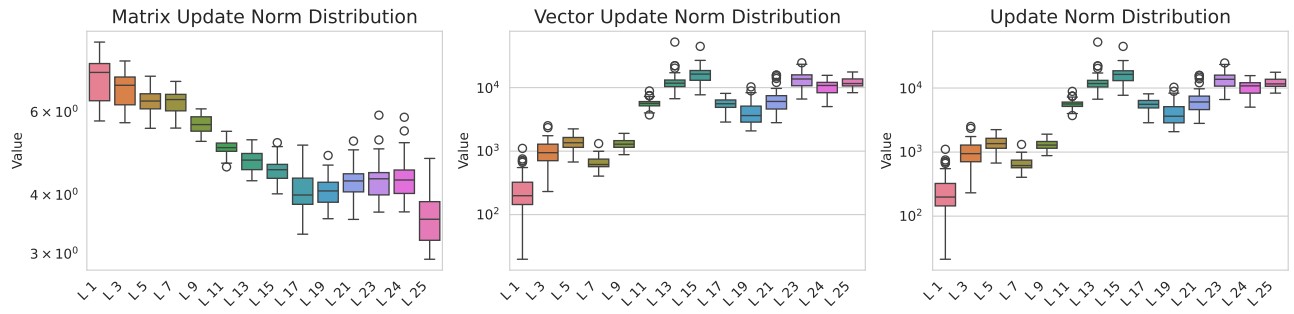

*(b)* `float32` / Stable

Summary Statistics Over All Generated Tokens

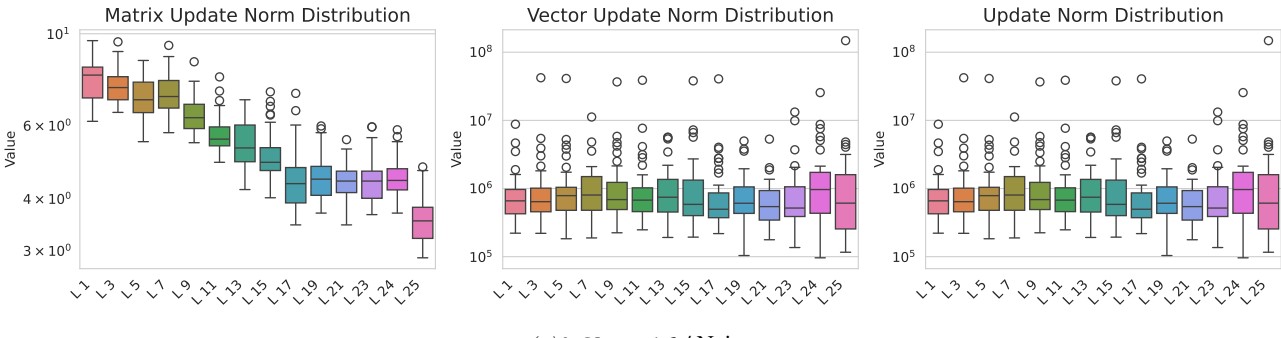

*(c)* `bfloat16` / Naive

Summary Statistics Over All Generated Tokens

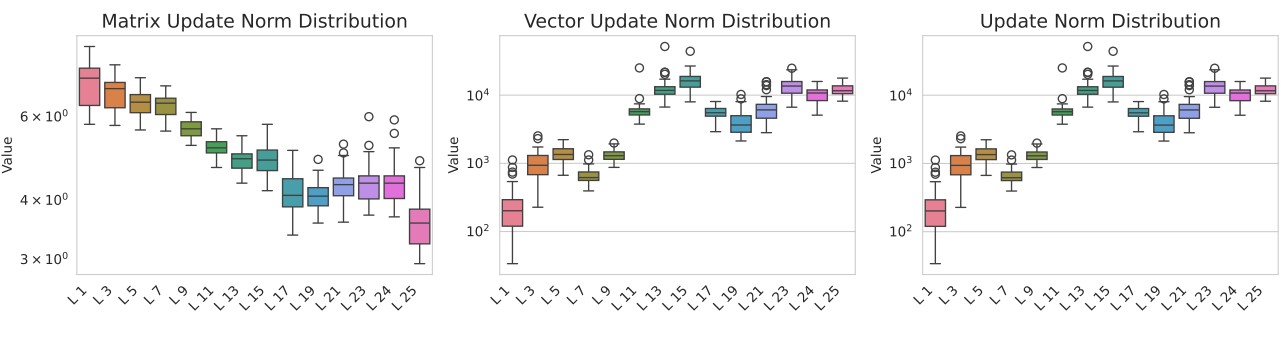

*(d)* `bfloat16` / Stable

*Figure 11.* Norms for different starting layers. The stable update produces a much smaller update which gets even smaller with an earlier starting layer. The original parameters have norm around $10^4$

Summary Statistics Over All Generated Tokens

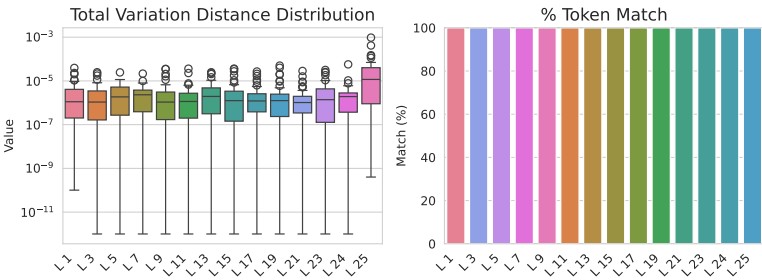

*(a)* `float32` / Naive

Summary Statistics Over All Generated Tokens

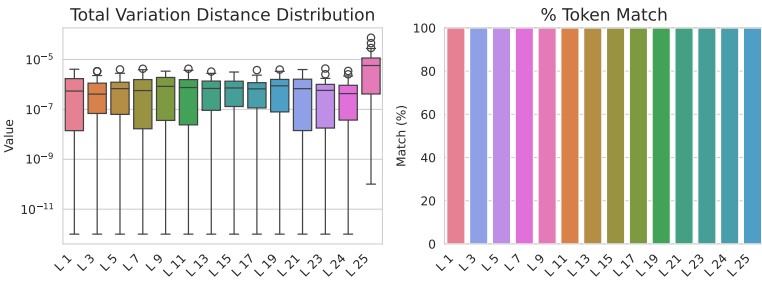

*(b)* `float32` / Stable

Summary Statistics Over All Generated Tokens

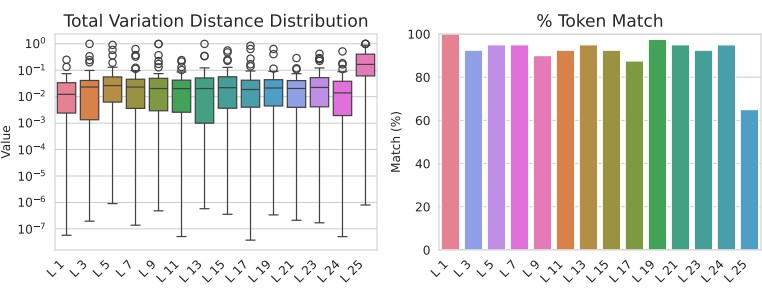

*(c)* `bfloat16` / Naive

Summary Statistics Over All Generated Tokens

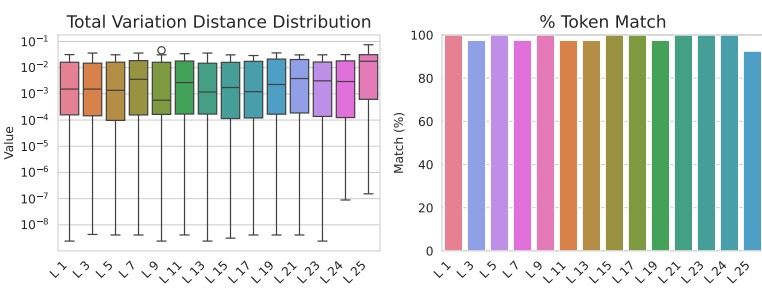

*(d)* `bfloat16` / Stable

*Figure 12.* Impact of the starting layer index on generation fidelity. Updating only the last layer decreases accuracy, however even a few layers are enough for good performance.

# Summary Statistics Over All Generated Tokens

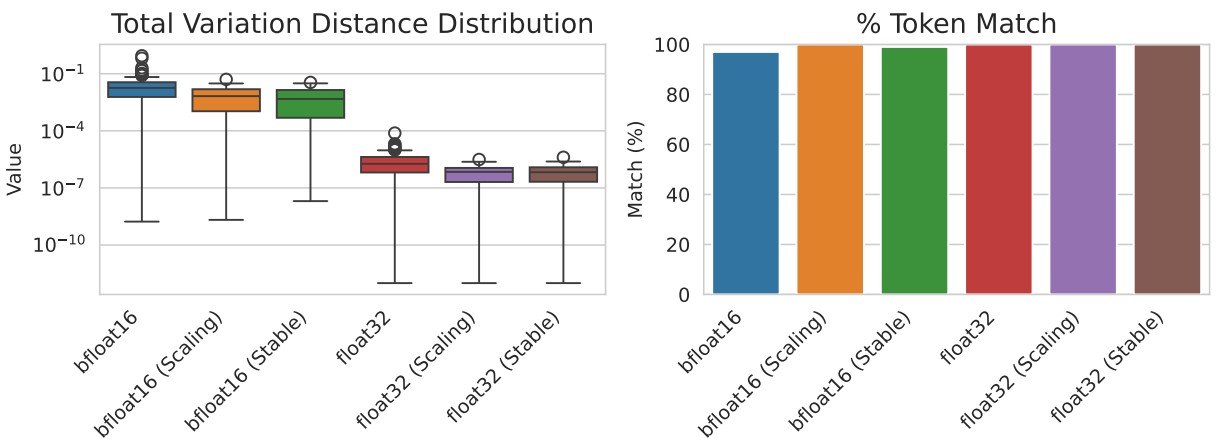

*(a)* Main Performance Metrics (TVD, Token Match)

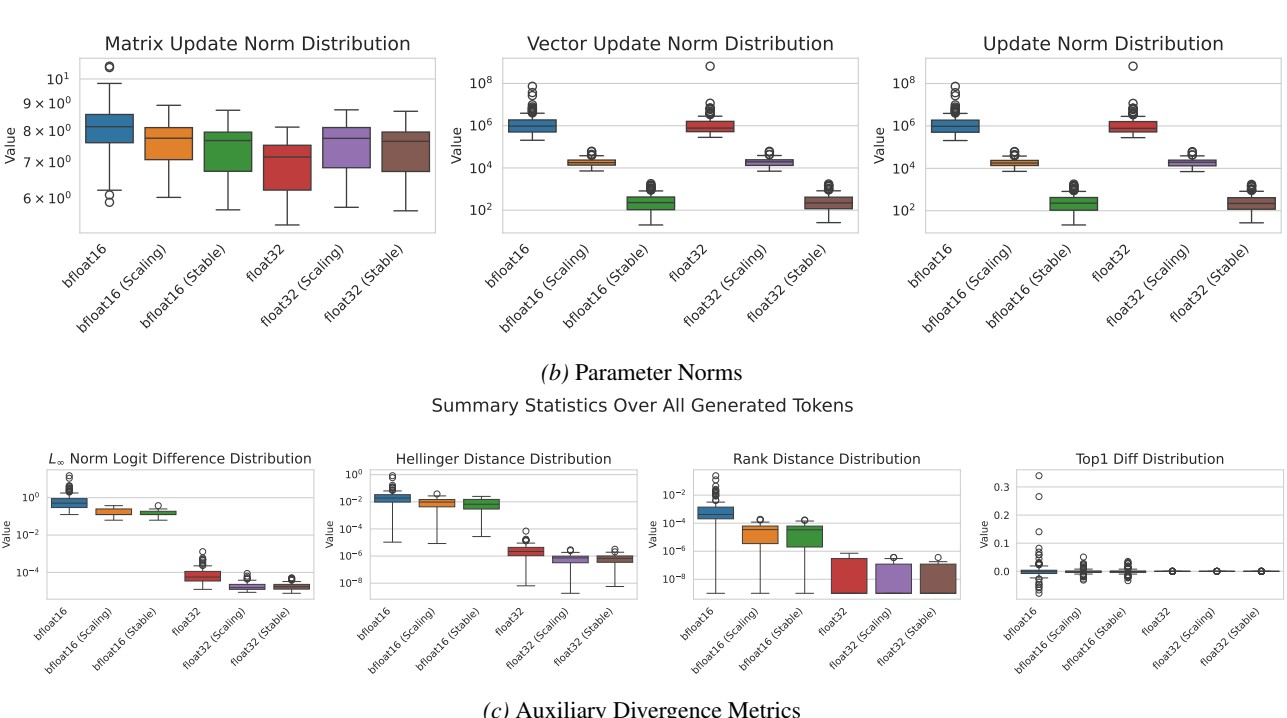

*(b)* Parameter Norms

*(c)* Auxiliary Divergence Metrics

*Figure 13.* Comprehensive evaluation of the scaling update variants compared to the naive and stable updates. The graphs include both `bfloat16` and `float32` to highlight numerical sensitivity. It can be seen that scaling and stable updates perform similarly, the scaling update is simpler while the stable update results in a much smaller norm update.

