# OpenReview forum: "Equivalence of Context and Parameter Updates in Modern Transformer Blocks"
_ICML.cc/2026/Conference — ICML 2026 spotlight_

### Official Review · Reviewer_zq5V · 2026-03-09

**Soundness:** 3
**Presentation:** 3
**Significance:** 2
**Originality:** 3
**Overall Recommendation:** 4
**Confidence:** 3

**Summary:**

This paper studies in-context learning through the lens of implicit re-parameterization. The main contribution is to show that, for more practical Transformer architectures including Gemma-style blocks, the effect of context can be represented as an equivalent parameter update inside the model. The paper further extends this view to multi-layer settings and supports the theory with experiments showing close agreement between the contextualized model and its re-parameterized counterpart.

**Compliance With Llm Reviewing Policy:**

Affirmed.

**Final Justification:**

Through the rebuttal, I raised my score.
I now understand the scope of this paper and acknowledge that just showing the existence (to show that the context can be compressed into weight difference) is valuable.

**Key Questions For Authors:**

- Does this theory truly reflect the difficulty of in-context learning? More specifically, can the formulation of the contextual effect as a single vector difference $z_c - z$ be further justified from the viewpoint of ICL?
- Is there a more explicit or constructive characterization of $z_c - z$ ? It seems that giving a more concrete formulation of this quantity may be essential for making the theory informative about ICL, rather than only about generic re-parameterization.
- Beyond validating the theory, is there any practical motivation for the experiments or any downstream application of this equivalence?
- What happens for architectures or settings in which no such re-parameterized expression exists for the in-context state? Would that imply a limitation of the framework, or could it reveal something important about which models can or cannot realize ICL in this form?

**Limitations:**

yes

**Strengths And Weaknesses:**

Strengths:
* The paper studies ICL as an equivalent re-parameterization of Transformers with more practical architectures, including Gemma-style blocks.
* The theory is extended beyond a simplified vanilla setting to more modern architectures, which is a meaningful technical contribution.
* The equivalence is supported by experiments.

Weaknesses:
* The paper does not sufficiently clarify the structure of ICL itself. In the main theorem, the effect of “in-context” information is essentially summarized by the difference $z_c -z$, i.e., the gap between two input vectors. As a result, the formulation appears too general to capture what is specific to ICL.
* Similarly, the difference between attention layers with and without context is treated in a rather compressed way, which makes it difficult to understand what aspect of attention is actually responsible for ICL behavior.
* I found the broader significance of the re-parameterization equivalence somewhat unclear. What does Theorem 1 tell us about the mechanism of Transformers in ICL, beyond an existence-type equivalence? For example, it would be interesting if this re-parameterization could be related to a more interpretable update rule, such as one-step gradient descent on some implicit objective.
* The implications of Definitions 1 and 2 for understanding the mechanism of ICL are not fully developed. While the definitions provide a general framework, the connection back to the actual difficulty and structure of ICL remains limited.
* **Impact Statement is lacked**.

---

> ### Author Rebuttal · Authors · 2026-03-30
>
> Thank you for your review. We are glad you found our paper a "meaningful technical contribution" and that our results are "supported by experiments".
>
> We hope our responses above address the substantive concerns and invite you to reconsider the overall score.
>
> **Weaknesses**
>
> The reviewer raises several points (weaknesses W1-4 and questions Q1-2) which stem from a shared theme: the desire for our framework to explain the specific learning dynamics of In-Context Learning (e.g., how attention retrieves tokens or how the update relates to SGD). We believe many of the concerns stem from a difference in scope rather than weaknesses in the paper itself, which we hope to clarify.
>
> As noted, our framework is very general and treats the attention mechanism as a black box looking only at the vector difference $z_C-z$. We view this as a strength rather than a limitation as it provides a tractable lens to inspect any ICL architecture.
>
> It is *not the goal of this paper* to propose a new explanation for *why* ICL works nor do we make the claim that ICL is literally making this update. Our contribution is proving that a prompt is *mathematically equivalent* to a specific weight update, and crucially, we prove this holds for real-world, complex architectures (e.g., Gemma, Llama, Falcon).
>
> While devising a complete theory of ICL is beyond the scope of this paper, our framework *can* help make progress towards understanding ICL. The fact that ‘the effect of in-context information is essentially summarized by the difference’ is a mathematical fact for single token prediction that lets us focus on a tractable part of the ICL update. $z_C-z$ carries the semantic content of whatever context Y contains and our framework allows one to inspect how that content can be distributed across layers and weight matrices, something not previously possible for modern architectures.
>
> This work also opens up numerous pathways for understanding ICL:
> * We can look at the matrix updates to see how a context impacts the weights, does the update occur more in specific layers? In specific parts of the prompt? Does it depend on what type of prompt it is?
> * How do the updates from this approach compare to SGD steps? Can we learn about the "learning" algorithm in ICL?
> However, we want to make it clear: It is *not the goal* of this work to better understand ICL, we only prove that we can always compress the context into weight updates at the token-level.
>
> Below, we address your other questions and clarify the core focus of our work
> >*[...] it would be interesting if this re-parameterization could be related to a more interpretable update rule, such as one-step gradient descent on some implicit objective*
>
> Our formulation builds on Dherin et al. (2025), where the update is the derivative of a specific trace-loss. This loss can be applied to $W_{gate}$ and $W_{in}$ across layers and adapted for $W_{down}$ and $m$. We will include this derivation in the final version.
>
> >*Impact Statement is lacked*
>
> Thank you for pointing this out, it has been fixed in the manuscript, apologies for the oversight.
>
> **Questions:**
> >*is there any practical motivation for the experiments or any downstream application of this equivalence?*
>
> Yes, the ability to compile context into weights unlocks several highly practical applications. In Section 2, we referenced a concurrent work (Mazzawi et al., 2025) that has begun exploring how to aggregate these into reusable, token-independent "thought patches." While out of scope for this paper, preliminary results combining the richer architectural updates derived here with the aggregation method proposed in Mazzawi et al. show that we can absorb non-trivial prompts into the weights.
>
> >*What happens for architectures [...] in which no such re-parameterized expression exists [...]? Would that imply a limitation [...] or could it reveal something important about which models can [...] realize ICL in this form?*
>
> The framework is very general and applies to all modern real-world architectures that we looked at. We can create pathological examples not used in practice where it would not always be possible, such as RMSNorm without a trainable scaling vector. We do not believe that ICL is intrinsically different in that setup, as we have stated we do not claim that this update is what ICL is doing, merely that prompts are equivalent to our proposed weight updates. Therefore, an architecture failing this re-parameterization would simply represent a limitation of our mathematical compilation tool for that specific edge case, rather than a claim about the intrinsic ICL capabilities of the architecture itself.
>
> **Presentation score**
>
> We note that a score of 2 for presentation was given without specific issues being raised, we would genuinely welcome concrete suggestions in the discussion phase so we can address them.
>
> We hope these clarifications show the paper's contributions are fully addressed within their stated scope.

---

> > ### Author Rebuttal · Reviewer_zq5V · 2026-04-01
> >
> > Thank you for your clear response.
> >
> > Overall, it seems that my concern was mainly due to the difference in the scope.
> > I found that the goal of this paper is to show the equivalence (in a practical setting) and it is valuable for some practical purposes.
> > It was also interesting that the shift of parameters can be represented as the gradient of some loss function.
> >
> > I will raise my overall score (and presentation score)

---

### Official Review · Reviewer_Eqzi · 2026-03-12

**Soundness:** 3
**Presentation:** 3
**Significance:** 3
**Originality:** 3
**Overall Recommendation:** 5
**Confidence:** 3

**Summary:**

The paper builds on a very recent preprint that shows how changes in the hidden state of an auto-regressive LLM (due to prior time steps) can also be formulated as weight changes applied to the original model.

While the prior work used the bias term in MLPs to absorb residual connections, these are not needed for the “weight-patch” used in this work.

The work shows how one can take the hidden state built up by regressing on $t$ timesteps and applying it as a weight change in a more production-appropriate setting than the prior work, for modern LLMs. This could be beneficial for baking in “prompts” to pre-trained models.

The work empirically confirms that the weight updates result in distributions similar to those with the prompts in at least the settings considered when using fp32 precision, while bf16 shows instability.

**Compliance With Llm Reviewing Policy:**

Affirmed.

**Final Justification:**

The author's gave a good rebuttal. I have adjusted the scores accordingly.

**Key Questions For Authors:**

Did the authors consider running the evaluations on more complex tasks? E.g., with longer predictions or a needle in a haystack from the prompt? I would like to know how if the distributions diverge over time. One can imagine the information in the patch having to be richer the more prior time steps it is meant to capture.

Since the evaluation relies on teacher forcing to evaluate distribution similarity, it would be good to see an evaluation without it. Perhaps on some task that uses prompts? It would be good to see how the errors compound in an applied setting.

Since instability is a major concern, do you have some thoughts on how the constrained inversion approximates the theoretical derivation?

**Limitations:**

The impact statement is missing.

**Strengths And Weaknesses:**

The idea that activations (the computations over time) can be replaced with post-hoc weight changes given the change in activations is mathematically clear up front and the idea in the prior work in that way makes sense. The paper is generally well written and the single-block Gemma proof is presented cleanly.

The paper reads somewhat as an engineering contribution in the sense that it targets specific set of models, and it provides one way to get mathematical equivalence on the output. This is not a weakness per se.The paper does introduce a general, albeit simple, theoretical framework of input and output controllability, which allows it to generalize and show it for multiple layers using induction.

While any parameterized equation will have a myriad of different ways to force some output, as the authors point out, it makes me wonder what is special about the formulations offered by the paper. Is there something meaningful about, e.g., the constrained RMSNorm inversion w.r.t. how the model computes and stores state, or is it an arbitrary change to force some output? I wonder if there are some desiderata for what would make one solution better than another, since there are so many degrees of freedom. Would an approach that is stackable (think multiple lora patches, or the cited work by Mazzawi) be a better solution than one that isn't, for instance?

Minor presentation issue: the bright green references are hard to read for me.

---

> ### Author Rebuttal · Authors · 2026-03-30
>
> Thanks so much for your review. We are very happy that you found the paper "well-written" and that the "single-block Gemma proof is presented cleanly". We hope the following addresses your concerns.
>
> **Weaknesses**
>
> > *Is there something meaningful about[...] how the model computes and stores state, or is it an arbitrary change to force some output? [are there] desiderata for what would make one solution better than another[...]. Would an approach that is stackable [...]be a better solution [...]?*
>
> The paper focuses on the existence rather than classifying which updates are "better", however the updates do try to be minimal in the following sense:
> * Rank-1 patches are in some sense the simplest updates we can make to a matrix and are easier to study for mechanistic interpretability
> * Spreading updates across layers ("stackability") reduces update norm and increases numerical stability (see Appendix C.3)
> * Matrix updates are more ‘specific’ than vector updates in terms of when they apply (a rank-1 update changes matrix outputs on only the portion of a vector input in that direction). This is the secondary logic of inverting the RMSNorm to minimize the update to m (beyond the numerical stability)
> We have added a note to the manuscript that summarises these points.
>
> If by "stackability" you instead mean the ability to combine multiple updates, then we agree that this is also a useful property for many applications, but one that goes beyond the single-token setup we use here.
>
> > *running the evaluations on more complex tasks? E.g., with longer predictions or a needle in a haystack from the prompt? I would like to know how if the distributions diverge over time*
>
> We did include an entire image as an input which is significantly more complex, however with the current single token setup we prove (teacher forcing + a new update for each generated token), this is always exact and does not depend on prompt length/complexity. This would only come into play for extensions where the weights are updated to absorb a prompt in such a way to allow multiple token generation, which is out of scope for this work.
>
> **Questions**
>
> > *Since the evaluation relies on teacher forcing to evaluate distribution similarity, it would be good to see an evaluation without it. Perhaps on some task that uses prompts?*
>
> This question and common theme in these questions point to the same underlying issue, which is worth clarifying directly. Our theoretical result is a statement about single-token equivalence: given any generation history, the patch perfectly replicates the next-token distribution. As such teacher forcing is not a limitation but the correct evaluation for this claim. The experiments with image context (Fig 5) demonstrate the approach scales to complex, high-dimensional prompts within this single-token framing. Evaluating on free generation would test a much stronger claim of a token-independent patch which we flag as future work. As discussed in our paper, Mazzawi et al (2025), take some steps towards this which can be combined with our improvements, along with ideas such as being careful about which layers to update and pushing the update into matrices.
>
> > *It would be good to see how the errors compound in an applied setting*
>
> We assume you are asking about errors compounding over layers not time, if you mean time then we refer to the answer just above.
> The algorithm was written in a slightly simplified form for readability, however it is actually self-correcting (we have updated the manuscript to make this more clear too). Each layer's update is calculated based on the actual (potentially noisy) output of the previous updated layer. Without this there are some errors that compound, especially if not using the stable method or if using bfloat16.
>
> > *do you have some thoughts on how the constrained inversion approximates the theoretical derivation?*
>
> Appendix B presents the exact solution to the minimization, there is no theoretical derivation that gets closer. To see how it compares to the goal we refer to Fig 8/9 which shows the norm of the update to m under different settings. Any error in the approximation for a block is also corrected in the next block.
>
> > *The impact statement is missing*
>
> We have added an impact statement to the updated draft.
>
> > *bright green references are hard to read*
>
> Agreed, this is fixed now.
>
> **Significance and originality scores**
>
> We note that you gave significance and originality scores of 2. The controllability framework is the key contribution: it reduces analysis of any future architecture to checking two simple conditions, rather than constructing new proofs case-by-case. This, along with numerically stable results, allows practical applications of this theory.
>
> We hope our clarifications, particularly the controllability counterexample and compilation discussion, address your remaining concerns and encourage you to raise your score.

---

> > ### Author Rebuttal · Reviewer_Eqzi · 2026-04-01
> >
> > Thanks for the good response. The authors have addressed my main concerns, and I will raise my score -- best of luck.

---

> > > ### Author Response · Authors · 2026-04-02
> > >
> > > Thank you for the encouraging response and for your decision to increase the score. We just wanted to gently note that the official score field currently remains unchanged, in case the update didn't save. Thank you again for your feedback.

---

### Official Review · Reviewer_szw9 · 2026-03-15

**Soundness:** 3
**Presentation:** 3
**Significance:** 3
**Originality:** 2
**Overall Recommendation:** 4
**Confidence:** 4

**Summary:**

The paper builds on previous work showing in-context learning updates in Transformers are equivalent to implicit low-rank weight updates in the MLP block, and generalizes this result across modern architectures and for multi-layer models. Going beyond theoretical contribution, the authors examine the validity of their result by testing whether they can dynamically update the weights of a Gemma model using an algorithm based on their equivalence theorems, such that it produces similar outputs across multi-token generation. They find that a no-context model with implicit weight updates according to their method nearly perfectly matches the outputs of a full-context model, and that quantization contributes to the success of their method. Finally, the authors generalize their theorems beyond the Gemma architecture.

**Compliance With Llm Reviewing Policy:**

Affirmed.

**Final Justification:**

Thank you to the authors for their engagement during the review period. As mentioned in my original review, I see this as a strong paper that is clear and well-presented, and has an interesting core result. Hence, I support its acceptance. As noted in my rebuttal acknowledgement, my concerns were mostly addressed (though I still believe additional results from other, more common architectures than Falcon, would be worthwhile for the camera ready), and I chose to maintain my positive score though not raise it given my view that the paper has somewhat sparse, though sufficient, empirical grounding.

**Key Questions For Authors:**

1. Given that theorem 3 generalizes the equivalence between ICL and implicit updates across architectures, could there be an algorithm to automatically compute the form of these updates (or whether they are possible) based on the architecture/computation graph rather than the form of required updates being derived manually? I would find this quite exciting if so.
2. Is there a way to implement this method without absorbing previously generated tokens at each time step? Meaning, could you provide the transformer an implicit update equivalent to some context and then let it generate freely, and it would keep behaving as if it sees this context? Or would the attention values for the new generated tokens disrupt this process?

**Limitations:**

Limitations are not (or minimally) addressed in the main text. Please include a limitations paragraph in your discussion if space permits.

**Strengths And Weaknesses:**

Strengths:
- The paper shows a strong result of generalizing the formal equivalence of in-context learning and implicit weight updates to modern architectures, as well as showing that this equivalence holds in practice through strong results that implicit updates can replicate the outputs of providing context in a no-context model.
- I see this as an important and exciting result that is likely to be of interest to the community, particularly since it could enable novel steering methods, as well as further investigation of the nature of parameteric versus in-context updates. The authors also provide a practical algorithm to implement the method in a multi-layer model.
- The paper is clear and well-written, the proofs and rationale were easy to follow, and the results are straightforward.

Weaknesses:
- The results provided are only for one architecture (Gemma), and while the authors claim to generalize theorems 1 and 2 across architectures in theorem 3, empirical results from an additional architecture would provide strong support for that claim.
- Novelty of the paper is not very strong, given that it directly builds on previous results by Dherin et al. (2025) by generalizing their claims across architectures. With that said, despite this contribution being somewhat incremental, extending these results to modern architectures is a crucial step for their practical significance.
- It is somewhat unclear to me to what extent the input controllability and output controllability constraints on theorem 3 are trivial or not. It would make the final section substantially clearer and make it seem less tautological if the authors would discuss what kinds of functions cannot be captured under these constraints (other than the simple 0 vector case).
- the empirical results provided in the paper are somewhat sparse, and the authors could consider adding results for different architectures, and perhaps show how these updates look in activation space, or contrast them with parametric updates through fine tuning.

Overall, I see this as a strong paper with interesting and clear results, which provides a valuable, though not highly novel, contribution. I think some revisions, particularly with respect to adding further empirical results, can improve the paper, but I think it is already a good contribution, and I recommend it should be accepted.

---

> ### Author Rebuttal · Authors · 2026-03-30
>
> Thank you for your review. We are delighted to hear that you found this “an important and exciting result that is likely to be of interest to the community”, and that you found the paper and proofs “easy to follow”.
>
> **Weaknesses**
>
> >*empirical results from an additional architecture would provide strong support for that claim*
>
> Most other architectures (e.g. Falcon, Llama, Mistral)  do not use post-norm, removing the need for the approximate RMSNorm inversion. This removes the most numerically unstable part of the update, making them easier in practice. While we do not have extensive experiments for other architectures, we list preliminary results from Falcon here. If accepted we will include more detailed results for Falcon.
>
> Falcon results using the same prompt from the paper (L2 norms of differences after layer residuals):
> Float32: all layer L2 diffs under 1.5E-4 (vs ~3E-3 for Gemma)
> Float16: all layer L2 diffs under 0.5 (vs ~5 for Gemma)
>
> >*Novelty of the paper is not very strong, given that it directly builds on previous results [...] generalizing their claims across architectures. [...]despite this [...] extending these results to modern architectures is a crucial step for their practical significance*
>
> Regarding our results novelty, we show that the implicit update phenomenon is universal across all modern architectures. We also provide a simple framework to extend to new architectures. Specifically the extension to architectures with no biases, post-norm and multiple layers is a rigorous theoretical step requiring a new conceptual machinery - our new input/output controllability framework. Furthermore, identifying and solving the severe numerical instability inherent to approximate RMSNorm inversion in bfloat16 is a highly non-trivial, novel contribution essential for practical application.
> >*the authors would discuss what kinds of functions cannot be captured under these constraints*
>
> If an architecture uses RMS post-norm without a trainable scaling vector m, output controllability is impossible. This can be seen as the output is strictly constrained to the L2 sphere; the model cannot stretch the vector to absorb a delta whose scale differs from the norm. No major production model uses this configuration, illustrating that the constraints are non-trivial. We will add this discussion in the final version.
> >*adding results for different architectures, and perhaps show how these updates look in activation space, or contrast them with parametric updates through fine tuning*
>
> While comparing these implicit updates to parametric fine-tuning or visualizing them in activation space is a fascinating direction for mechanistic interpretability, it falls outside the scope of this paper; our updates are rank-1 by construction while fine-tuning produces dense updates, making direct comparison non-trivial and worthy of its own investigation.
>
> To address the request for broader empirical validation, we have additionally run our algorithm on Falcon (see above).
> >*could there be an algorithm to automatically compute the form of these updates (or whether they are possible) based on the architecture/computation graph*
>
> We agree this is a highly exciting direction! Thm 3 establishes the mathematical foundation required for such an auto-compiler. By traversing the computational graph, one could automatically track values with/without context and apply controllability updates whenever there is an operation where it is valid. Because the solution is not always unique, the 'best' automated update would depend on specific use-case constraints (e.g., minimizing numerical drift, or pushing as much of an update as possible into matrix updates). We will add a mention of this automatic-compilation vision to the paper.
> >*Is there a way to implement this method without absorbing previously generated tokens at each time step?*
>
> Yes, this is possible. As we briefly touch upon in Section 2, concurrent work (Mazzawi et al., 2025) has begun exploring how to aggregate these into reusable, token-independent 'thought patches.' As you intuited, naive implementation is disrupted by new attention values. Making these patches more robust requires solving several out-of-scope challenges to ensure the patch generalizes across queries, such as:
> 1. aggregating updates across multiple generation steps
> 2. carefully choosing which layers to update (e.g. avoiding first/last layer)
> 3. relying strictly on numerically stable updates (avoiding updating m)
>
> >*[...] Please include a limitations paragraph in your discussion if space permits*
>
> Thank you for pointing this out. While we are tight on space, we will make space for a Limitations paragraph in the discussion section addressing 1) the token-dependent nature of the strict mathematical equivalence, and 2) the limits of the gained insights into ICL
>
> We hope the inclusion of the new Falcon results and our clarifications address your remaining concerns and encourage you to raise your score if you agree.

---

> > ### Author Rebuttal · Reviewer_szw9 · 2026-04-04
> >
> > Thank you to the authors for their response. My concerns have been mostly addressed (though I still believe additional results from other, more common architectures than Falcon, would be worthwhile). However, I think my (already positive) score reflects my current view of the paper: A strong and interesting theoretical and empirical result with interesting potential future directions, but somewhat sparse, though sufficient, empirical grounding. Hence, I will maintain my current score.

---

### Decision · Program_Chairs · 2026-04-30

**Decision:**

Accept (spotlight)

**Comment:**

This paper establishes a solid theoretical and practical result: the effect of in-context information in modern transformer architectures can be exactly represented as implicit, low-rank weight updates to MLP blocks. Building on prior work in simplified settings, the authors rigorously extend the equivalence to realistic architectures (e.g., Gemma-style blocks, gating, pre/post-norm, multi-layer models) via a clear input/output controllability framework, and provide constructive algorithms.
The contribution is well presented and technically sound. Reviewers consistently find the proofs clear and the theory compelling, and the empirical results convincingly demonstrate matching between contextualized models and their re-parameterized counterparts. While the empirical evaluation is relatively limited in architectural breadth, preliminary results on additional models and the authors’ willingness to add limitations and clarification mitigate this concern.

Overall, this is a valuable theory paper with solid empirical validation and clear implications for understanding and manipulating in-context learning. I recommend Accept.